# Interleukin-1 prevents SARS-CoV-2-induced membrane fusion to restrict viral transmission via induction of actin bundles

Xu Zheng[1†], Shi Yu[1†‡], Yanqiu Zhou[2†], Kuai Yu[3†], Yuhui Gao[1], Mengdan Chen[1], Dong Duan[1,4], Yunyi Li[2], Xiaoxian Cui[2], Jiabin Mou[2], Yuying Yang[2], Xun Wang[5], Min Chen[2]*, Yaming Jiu[1]*, Jincun Zhao[3]*, Guangxun Meng[1,4]*

[1]The Center for Microbes, Development and Health, National Key Laboratory of Immune Response and Immunotherapy, Shanghai Institute of Immunity and Infection, Chinese Academy of Sciences, University of Chinese Academy of Sciences, Shanghai, China; [2]Shanghai Municipal Center for Disease Control and Prevention, Shanghai, China; [3]The First Affiliated Hospital of Guangzhou Medical University, State Key Laboratory of Respiratory Disease, National Clinical Research Center for Respiratory Disease, Guangzhou Institute of Respiratory Health, Guangzhou, China; [4]School of Life Sciences, Soochow University, Jiangsu, China; [5]Shanghai Blood Center, Shanghai, China

*For correspondence:
chenmin@scdc.sh.cn (MC);
ymjiu@siii.cas.cn (YJ);
zhaojincun@gird.cn (JZ);
gxmeng@ips.ac.cn (GM)

†These authors contributed equally to this work

Present address: ‡Guangzhou National Laboratory, Guangzhou International Bio Island, Guangdong, China

**Competing interest:** The authors declare that no competing interests exist.

## eLife Assessment

This study provides **important** insights into how IL-1 cytokines protect cells against SARS-CoV-2 infection. By inducing a non-canonical RhoA/ROCK signaling pathway, IL-1beta inhibits the ability of SARS-CoV-2 infected cells to fuse with uninfected cells and produce syncytia. **Convincing** evidence underlies the identification of the key signaling components required for this inhibitory phenotype, and suggests that this process may also function to inhibit SARS-CoV-2 infection in vivo.

## Abstract

Innate immune responses triggered by severe acute respiratory syndrome coronavirus 2 (SARS-CoV-2) infection play pivotal roles in the pathogenesis of COVID-19, while host factors including proinflammatory cytokines are critical for viral containment. By utilizing quantitative and qualitative models, we discovered that soluble factors secreted by human monocytes potently inhibit SARS-CoV-2-induced cell-cell fusion in viral-infected cells. Through cytokine screening, we identified that interleukin-1β (IL-1β), a key mediator of inflammation, inhibits syncytia formation mediated by various SARS-CoV-2 strains. Mechanistically, IL-1β activates RhoA/ROCK signaling through a non-canonical IL-1 receptor-dependent pathway, which drives the enrichment of actin bundles at the cell-cell junctions, thus prevents syncytia formation. Notably, in vivo infection experiments in mice confirmed that IL-1β significantly restricted SARS-CoV-2 spread in the lung epithelium. Together, by revealing the function and underlying mechanism of IL-1β on SARS-CoV-2-induced cell-cell fusion, our study highlights an unprecedented antiviral function for cytokines during viral infection.

## Introduction

The COVID-19 pandemic caused by severe acute respiratory syndrome coronavirus 2 (SARS-CoV-2) infection has spread globally, with at least 755 million people diagnosed and the death toll is over 6.8 million. SARS-CoV-2 variants of concern, including Alpha, Beta, Delta, and Omicron, continue to

**eLife digest** SARS-CoV-2, the agent responsible for COVID-19, has claimed millions of lives across the globe. To better manage this disease and develop new treatments, it is fundamental to understand how the immune system responds to this virus – and, in particular, how it can be thwarted.

Like all viruses, SARS-CoV-2 replicates within host cells and bursts out when it has made more of itself and is ready to infect more tissues. It can also cause neighbouring cells to merge, allowing the virus to replicate and spread without stepping outside. This strategy makes it harder for the immune system to access and deactivate the threat.

A group of molecules called proinflammatory cytokines (such as IL-1β and IL-1α) are released upon SARS-CoV-2 infection. People receiving immunosuppressive therapies, which can reduce proinflammatory cytokine levels to harness inflammatory damage, find it harder to tackle the virus. However, the full role of these molecules in clearing SARS-CoV-2 remains unknown.

To investigate this question, Zheng, Yu, Zhou and Yu et al. developed different experimental models that could examine how proinflammatory cytokines might protect cells from SARS-CoV-2 challenge. The results showed that IL-1β and IL-1α stop the virus from being able to fuse cells together. Further cell studies revealed the underlying mechanism: IL-1β triggers cells to increase the levels of essential components, known as actin bundles, which form the structures that prevent cells from fusing with each other. Experiments in live mice showed that IL-1β treatment significantly prevented SARS-CoV-2 from spreading within the lining of the lungs. Taken together, these findings reveal new insights into how the immune system protects hosts against SARS-CoV-2 infection; further investigation may help identify new treatments for COVID-19.

evolve and increase transmissibility and the ability to escape host immune responses. These variants have presented significant challenges to the design and development of vaccines and therapeutic agents (*Zhou et al., 2020*). In order to discover novel strategies to control the virus, it is important to understand host responses to SARS-CoV-2 infection.

SARS-CoV-2 infection induces cell-cell fusion (also known as syncytia formation) in multiple cell types including lung epithelial cells, neurons, and glia (*Martínez-Mármol et al., 2023*). Syncytia formation between SARS-CoV-2-infected cells with neighboring cells may contribute to increased viral transmission and pathogenicity in the infected host (*Rajah et al., 2022*), which also makes the virus insensitive to extracellular neutralizing antibodies (*Li et al., 2022*; *Yu et al., 2023*). Moreover, syncytia formation among pneumocytes with long-term persistence of viral RNA has been observed in the lung autopsy of deceased COVID-19 donors, which may contribute to prolonged clearance of the virus and long COVID symptoms (*Bussani et al., 2020*). Therefore, inhibiting syncytia formation is critical to ensure viral clearance and to control viral transmission.

It has been reported that SARS-CoV-2-mediated syncytia formation is effectively inhibited by multiple interferon (IFN)-stimulating genes (ISGs) (*Pfaender et al., 2020*; *Wang et al., 2020*). However, low IFN levels with impaired ISG responses were observed during early SARS-CoV-2 infection, which may have compromised the antiviral responses of IFN in severe COVID-19 patients (*Blanco-Melo et al., 2020*; *Hadjadj et al., 2020*). Thus, identifying other endogenous host factors that regulate syncytia formation is of great significance for harnessing the transmission of SARS-CoV-2.

Of note, a variety of cells are involved in the host responses to SARS-CoV-2 infection (*Ren et al., 2021*). Lung epithelial cells are the primary target of SARS-CoV-2 infection and transmission, which subsequently recruit and activate innate immune cells, leading to COVID-19 pathology (*Barnett et al., 2023*). In this process, tissue-resident macrophages and circulating monocytes contribute to local and systemic inflammation primarily by releasing inflammatory cytokines (*Sefik et al., 2022*). Among these cytokines induced by SARS-CoV-2, the combination of TNF-α and IFN-γ induces inflammatory cell death, resulting in clear tissue damage, while other cytokines' function remains obscure (*Karki et al., 2021*). In addition, when corticosteroids were applied to suppress the inflammatory response in patients infected by SARS-CoV (*Lee et al., 2004*) or MERS-CoV (*Arabi et al., 2018*), the clearance of viral RNA was obviously delayed, suggesting the importance of innate immune factors in viral clearance.

Innate immune cells express Toll-like receptors (TLRs), and TLR-mediated signaling induces robust production of inflammatory cytokines (*Medzhitov, 2001*). In the current work, we screened the role of soluble proinflammatory cytokines on the SARS-CoV-2 spike-induced cell-cell fusion. Notably, we identified that IL-1β, which is the key factor of inflammatory response, inhibited SARS-CoV-2 spike-induced syncytia formation in various cells by activating RhoA/ROCK pathway to initiate actin bundle formation at cell-cell interface between SARS-CoV-2-infected cells and neighboring cells. Importantly, IL-1β significantly reduced SARS-CoV-2 transmission among lung epithelia in experimental mice in vivo. Therefore, our data highlight an important role for proinflammatory cytokines against viral infection.

## Results

### Host factors secreted by activated innate immune cells inhibit SARS-CoV-2-induced cell-cell fusion

We have previously established quantitative and qualitative models for SARS-CoV-2 spike-induced cell-cell fusion by bioluminescence assay, immunoblotting, and fluorescence imaging (*Yu et al., 2022*). In order to explore the potential effect of cytokines on SARS-CoV-2-induced cell-cell fusion, human monocyte cell line THP-1 and human peripheral blood mononuclear cells (PBMCs) were used in this study. We applied several TLR ligands to stimulate such innate immune cells and collected the cell culture supernatants for subsequent experiments (*Figure 1A*). Of note, cell culture supernatants of THP-1 cells stimulated by TLR ligands significantly reduced the bioluminescence signal, while neither untreated THP-1 cell culture supernatant nor the medium control had any effect on the bioluminescence signal reflecting cell-cell fusion (*Figure 1B*). SARS-CoV-2 spike engagement of ACE2 primed the cleavage of S2′ fragment in target cells, a key proteolytic event coupled with spike-mediated membrane fusion (*Yu et al., 2022*). In parallel with bioluminescence assay, a large amount of enriched S2′ cleavage was detected in HEK293T-Spike and HEK293T-ACE2 co-cultured group and co-culture incubated with untreated THP-1 cell culture supernatant, while S2′ cleavage was clearly reduced upon treatment with TLR ligands-stimulated THP-1 cell culture supernatants (*Figure 1C*). Syncytia formation was also visualized using cells co-expressing spike and a ZsGreen fluorescent reporter. The area of syncytium were significantly reduced by the treatment with TLR ligands-stimulated THP-1 cell culture supernatants (*Figure 1D* and *Figure 1—figure supplement 1A*). Considering the presence of TLR ligands in such cell culture supernatants, we tested their potential direct effects. As expected, TLR ligands alone did not reduce the bioluminescence signal and S2′ cleavage compared to the control groups, as well as no effect on syncytia formation (*Figure 1—figure supplement 1B–E*).

Concurrently, we also tested the effect of PBMCs culture supernatants on SARS-CoV-2 spike-induced cell-cell fusion. Consistent with the results from THP-1 cells, TLR ligands-stimulated PBMCs culture supernatants treatment also strongly reduced the bioluminescence signal, S2′ cleavage, and the area of syncytium compared with the medium group (*Figure 1—figure supplement 1F–I*). These results thus suggested that activated innate immune cells released host factors to inhibit SARS-CoV-2 spike-induced cell-cell fusion.

To validate the effect of innate immune cell culture supernatants on cell-cell fusion in authentic SARS-CoV-2 infection, we pre-treated ACE2-expressing cells with THP-1 cell culture supernatants before inoculation with SARS-CoV-2 B.1.617.2 (Delta) or wild-type (WT) strains. Cell lysates were used for the detection of SARS-CoV-2 spike and N protein 24 hr post-infection (hpi) (*Figure 1E*). Result from this experiment showed that TLR ligands-stimulated-THP-1 cell culture supernatants reduced S2′ cleavage and N protein levels during Delta or WT SARS-CoV-2 infection in HEK293T-ACE2 cells, whereas untreated THP-1 cell culture supernatant had no effect (*Figure 1F*, *Figure 1—figure supplement 2A*). In addition, TLR ligands-stimulated-THP-1 cell culture supernatants reduced the area of syncytium induced by Delta or WT SARS-CoV-2 infection (*Figure 1—figure supplement 2B and C*). Furthermore, we infected the human colon epithelial carcinoma cell line Caco-2 with WT SARS-CoV-2, and found that S2′ cleavage and N protein amounts were reduced after TLR ligands-stimulated THP-1 cell culture supernatants pre-treatment (*Figure 1G*). Accordingly, immunofluorescent staining also showed that TLR ligands-stimulated THP-1 cell culture supernatants significantly reduced the area of syncytium during SARS-CoV-2 infection in Caco-2 cells (*Figure 1H and I*). Therefore, these data suggested that host factors secreted by activated innate immune cells inhibit authentic SARS-CoV-2-induced cell-cell fusion.

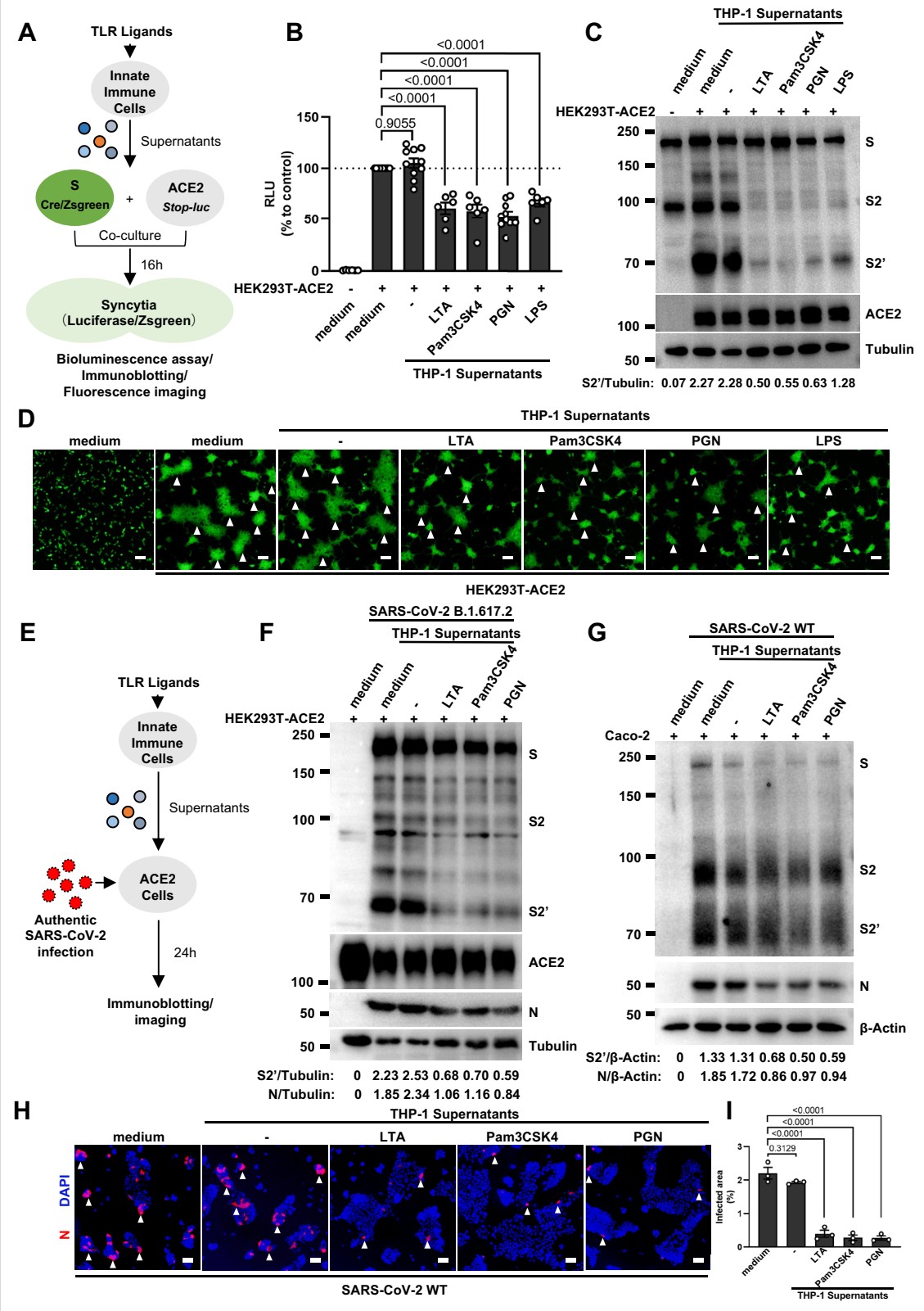

**Figure 1.** Host factors secreted by activated innate immune cells inhibit severe acute respiratory syndrome coronavirus 2 (SARS-CoV-2)-induced cell-cell fusion. (**A**) Schematics of the cell-cell fusion model used to quantify spike-mediated syncytium formation upon treatment with cell culture supernatants from Toll-like receptor (TLR) ligands-stimulated innate immune cells. Cells co-expressing SARS-CoV-2 spike and Cre were co-cultured with ACE2 and *Stop-luc* co-expressing HEK293T cells for 16 hr, before cell lysates were collected for bioluminescence assay and immunoblotting. Cells co-expressing

*Figure 1 continued on next page*

*Figure 1 continued*

SARS-CoV-2 spike and ZsGreen were co-cultured with ACE2 expressing HEK293T cells for 16 hr before fluorescence imaging. (**B**) Luciferase activity (relative luminescence units [RLU]) measured from HEK293T cell lysates collected from THP-1 supernatants-treated HEK293T-S and HEK293T-ACE2 described in (A) for 16 hr. Fetal bovine serum (FBS)-free RPMI 1640 served as medium control. Data are representative of six individual repeats and displayed as individual points with mean ± standard error of the mean (SEM). (**C**) Immunoblots showing full-length spike, S2, cleaved S2', and ACE2 collected from THP-1 supernatants-treated HEK293T-S and HEK293T-ACE2 described in (A) for 16 hr. Blots are representative of three independent experiments. Numbers below the blots indicated the intensity of S2' versus Tubulin. (**D**) Representative fluorescent image captured at 488 nm from THP-1 supernatants-treated HEK293T-S-ZsGreen and HEK293T-ACE2 for 16 hr. (**E**) Schematic presentation of THP-1 supernatants pre-treatment on authentic SARS-CoV-2-infected cells. Pre-treatment of HEK293T-ACE2 cells with THP-1 supernatants for 1 hr, then inoculated with 0.5 multiplicity of infection (MOI) Delta or wild-type (WT) authentic SARS-CoV-2 virus. Imaging was performed at 24 hr post-infection (hpi) before cell lysates were harvested for immunoblotting. (**F**) Immunoblots of Delta SARS-CoV-2 S, S2, cleaved S2', N, and ACE2 proteins collected from HEK293T-ACE2 cells 24 hpi as described in (E). Blots are representative of three individual experiments. Numbers below the blots indicated the intensity of S2' or N versus Tubulin. (**G**) Immunoblots of WT SARS-CoV-2 S, S2, cleaved S2', and N proteins collected from Caco-2 cells 24 hpi as described in (E). Blots are representative of three individual experiments. Numbers below the blots indicated the intensity of S2' or N versus β-Actin. (**H**) Immunofluorescent images showing morphology of SARS-CoV-2-infected Caco-2 cells pre-treated with THP-1 supernatants. Anti-SARS-CoV-2 N was stained with Alexa Fluor 555, and nuclei were counterstained with DAPI, respectively. White arrowheads (D and H) indicate syncytia formation or infected cells, scale bars are indicative of 50 μm, and images are representative of three independent experiments. (**I**) Quantification of the infected area in (H).

The online version of this article includes the following source data and figure supplement(s) for figure 1:

**Source data 1.** Annotated, uncropped blots in *Figure 1*.

**Source data 2.** Raw, uncropped blots in *Figure 1*.

**Source data 3.** Source data of the individual points in *Figure 1*.

**Figure supplement 1.** Host factors secreted by activated peripheral blood mononuclear cells (PBMCs) inhibit severe acute respiratory syndrome coronavirus 2 (SARS-CoV-2) spike-induced cell-cell fusion, but Toll-like receptors (TLR) ligands alone have no effect.

**Figure supplement 1—source data 1.** Annotated, uncropped blots in *Figure 1—figure supplement 1*.

**Figure supplement 1—source data 2.** Raw, uncropped blots in *Figure 1—figure supplement 1*.

**Figure supplement 1—source data 3.** Source data of the individual points in *Figure 1—figure supplement 1*.

**Figure supplement 2.** Host factors secreted by activated THP-1 cells inhibit authentic severe acute respiratory syndrome coronavirus 2 (SARS-CoV-2)-induced cell-cell fusion.

**Figure supplement 2—source data 1.** Annotated, uncropped blots in *Figure 1—figure supplement 2*.

**Figure supplement 2—source data 2.** Raw, uncropped blots in *Figure 1—figure supplement 2*.

## IL-1β inhibits SARS-CoV-2-induced cell-cell fusion

To explore which host factor(s) inhibited SARS-CoV-2-induced cell-cell fusion, we first detected mRNA levels of different cytokines in THP-1 cells stimulated by TLR ligands. It was found that the expression levels of *IL1A*, *IL1B*, *IL6*, and *IL8* were significantly increased upon TLR ligands stimulation, while *IL4*, *IL12A*, *IFNA1*, *IFNB1*, and *IFNG* mRNA levels were not changed or undetected (*Figure 2—figure supplement 1A*). In addition, we also detected the mRNA levels of cytokine receptors in HEK293T modeling cells, confirming that *IL1R1*, *IL4R*, *IL6ST*, *IL8RA*, *IFNAR1*, *IFNGR1* were expressed in such cells, while *IL2RA* and *IL12RB1* were undetectable (*Figure 2—figure supplement 1B*).

We next selected recombinant IL-1α, IL-1β, IL-6, and IL-8 to test whether individual cytokine may play a role in affecting SARS-CoV-2 spike-induced cell-cell fusion (*Figure 2A*). Interestingly, IL-1α and IL-1β significantly reduced the bioluminescence signal compared to the control group, while IL-6 and IL-8 had little or no effect (*Figure 2B*). In addition, fluorescence images of cells expressing ZsGreen reporter also confirmed that IL-1α and IL-1β significantly inhibited SARS-CoV-2 spike-induced syncytia formation (*Figure 2—figure supplement 1C and D*). Furthermore, IL-1β and IL-1α both reduced the bioluminescence signal and S2' cleavage (*Figure 2C and D* and *Figure 2—figure supplement 2A*) in cell lysates in a dose-dependent manner. Moreover, the syncytia formation was inhibited with increasing concentrations of IL-1β or IL-1α (*Figure 2—figure supplement 2B–E*). Intriguingly, when we added both IL-1α and IL-1β, there was no synergistic inhibition on cell-cell fusion compared to either cytokine alone (*Figure 2—figure supplement 2F–H*), suggesting a saturation of IL-1 receptor binding to these homologues. Since both IL-1α and IL-1β activate the downstream pathway through the same receptor IL-1R1, these data suggested that IL-1α or IL-1β may inhibit cell-cell fusion through the same pathway. Considering the higher mRNA level of *IL1B* than *IL1A*, as well as the classical

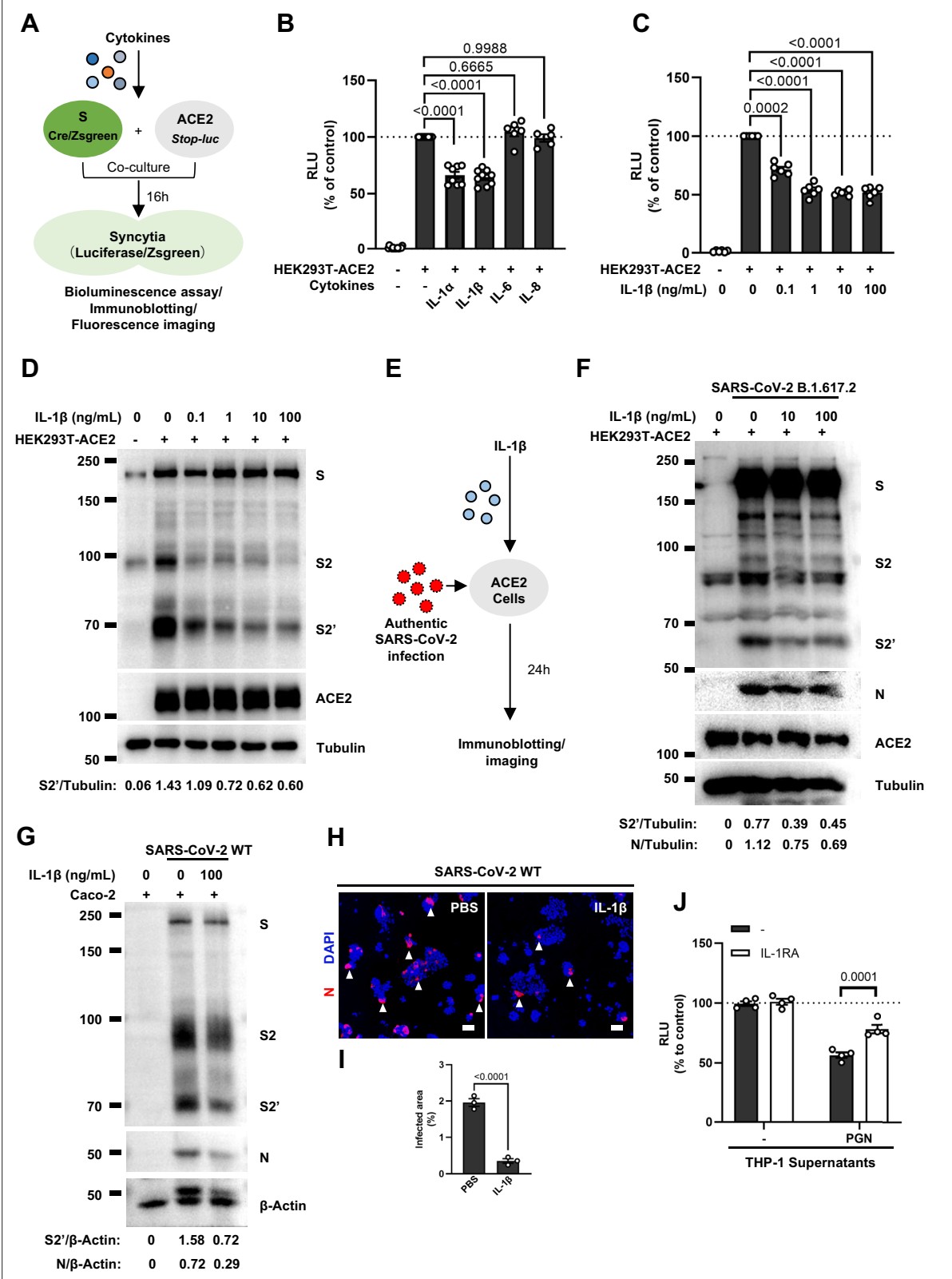

**Figure 2.** Interleukin-1β (IL-1β) inhibits severe acute respiratory syndrome coronavirus 2 (SARS-CoV-2)-induced cell-cell fusion. (**A**) Schematics of the cell-cell fusion model used to quantify spike-mediated syncytium formation upon treatment with different cytokines. Cells co-expressing SARS-CoV-2 spike and Cre were co-cultured with ACE2 and *Stop-luc* co-expressing HEK293T cells for 16 hr, before cell lysates were collected for bioluminescence assay and immunoblotting. Cells co-expressing SARS-CoV-2 spike and ZsGreen were co-cultured with ACE2 expressing HEK293T cells for 16 hr

*Figure 2 continued on next page*

*Figure 2 continued*

before fluorescence imaging. (**B**) Luciferase activity (relative luminescence units [RLU]) measured from HEK293T cell lysates collected from different cytokines-treated HEK293T-S and HEK293T-ACE2 described in (A) for 16 hr. IL-1α (10 ng/mL), IL-1β (1 ng/mL), IL-6 (100 ng/mL), or IL-8 (100 ng/mL) were added into the cell-cell fusion system. Data are representative of six individual repeats and displayed as individual points with mean ± standard error of mean (SEM). (**C**) Luciferase activity (RLU) measured from HEK293T cell lysates collected from different concentrations of IL-1β-treated HEK293T-S and HEK293T-ACE2 for 16 hr. Data are representative of six individual repeats and displayed as individual points with mean ± SEM. (**D**) Immunoblots showing full-length spike, S2, cleaved S2′, and ACE2 collected from different concentrations of IL-1β-treated HEK293T-S and HEK293T-ACE2 for 16 hr. Blots are representative of three independent experiments. Numbers below the blots indicated the intensity of S2′ versus Tubulin. (**E**) Schematic presentation of IL-1β pre-treatment on authentic SARS-CoV-2-infected cells. Pre-treatment of HEK293T-ACE2 cells with different concentrations of IL-1β for 1 hr, then inoculated with 0.5 multiplicity of infection (MOI) Delta or wild-type (WT) authentic SARS-CoV-2 virus. Bright-field images were captured at 24 hr post-infection (hpi) before cell lysates were harvested for immunoblotting. (**F**) Immunoblots of Delta SARS-CoV-2 S, S2, cleaved S2′, N, and ACE2 proteins collected from HEK293T-ACE2 cells 24 hpi as described in (E). Blots are representative of three individual experiments. Numbers below the blots indicated the intensity of S2′ or N versus Tubulin. (**G**) Immunoblots of WT SARS-CoV-2 S, S2, cleaved S2′, and N proteins collected from Caco-2 cells 24 hpi as described in (E). Blots are representative of three individual experiments. Numbers below the blots indicated the intensity of S2′ or N versus β-Actin. (**H**) Immunofluorescent images showing morphology of SARS-CoV-2-infected Caco-2 cells pre-treated with or without IL-1β. Anti-SARS-CoV-2 N was stained with Alexa Fluor 555, and nuclei were counterstained with DAPI, respectively. White arrowheads indicate syncytia formation or infected cells, scale bars are indicative of 50 μm and images are representative of three independent experiments. (**I**) Quantification of the infected area in (H). (**J**) Luciferase activity (RLU) measured from THP-1 supernatants-treated HEK293T-S and HEK293T-ACE2 in the presence or absence of IL-1 receptor antagonist (IL-1RA). Data are representative of four individual repeats and displayed as individual points with mean ± SEM.

The online version of this article includes the following source data and figure supplement(s) for figure 2:

**Source data 1.** Annotated, uncropped blots in *Figure 2*.

**Source data 2.** Raw, uncropped blots in *Figure 2*.

**Source data 3.** Source data of the individual points in *Figure 2*.

**Figure supplement 1.** mRNA levels of different cytokine genes in THP-1 cells and selected cytokine receptor genes in HEK293T cells, as well as fluorescent images indicating cell-cell fusion in the presence of indicated cytokines.

**Figure supplement 1—source data 1.** Source data of the individual points in *Figure 2—figure supplement 1*.

**Figure supplement 2.** No synergistic inhibition of severe acute respiratory syndrome coronavirus 2 (SARS-CoV-2) spike-induced cell-cell fusion by interleukin-1α (IL-1α) and IL-1β co-treatment.

**Figure supplement 2—source data 1.** Annotated, uncropped blots in *Figure 2—figure supplement 2*.

**Figure supplement 2—source data 2.** Raw, uncropped blots in *Figure 2—figure supplement 2*.

**Figure supplement 2—source data 3.** Source data of the individual points in *Figure 2—figure supplement 2*.

**Figure supplement 3.** Interleukin-1β (IL-1β) inhibits authentic severe acute respiratory syndrome coronavirus 2 (SARS-CoV-2)-induced cell-cell fusion.

**Figure supplement 3—source data 1.** Annotated, uncropped blots in *Figure 2—figure supplement 3*.

**Figure supplement 3—source data 2.** Raw, uncropped blots in *Figure 2—figure supplement 3*.

**Figure supplement 4.** Interleukin-1β (IL-1β) is an important host factor from innate immune cells inhibiting severe acute respiratory syndrome coronavirus 2 (SARS-CoV-2) spike-induced cell-cell fusion.

**Figure supplement 4—source data 1.** Annotated, uncropped blots in *Figure 2—figure supplement 4*.

**Figure supplement 4—source data 2.** Raw, uncropped blots in *Figure 2—figure supplement 4*.

**Figure supplement 4—source data 3.** Source data of the individual points in *Figure 2—figure supplement 4*.

**Figure supplement 5.** Interleukin-1β (IL-1β) inhibits severe acute respiratory syndrome coronavirus 2 (SARS-CoV-2)-induced cell-cell fusion through acting on both donor and acceptor cells.

**Figure supplement 5—source data 1.** Annotated, uncropped blots in *Figure 2—figure supplement 5*.

**Figure supplement 5—source data 2.** Raw, uncropped blots in *Figure 2—figure supplement 5*.

**Figure supplement 5—source data 3.** Source data of the individual points in *Figure 2—figure supplement 5*.

**Figure supplement 6.** Interleukin-1β (IL-1β) inhibits severe acute respiratory syndrome coronavirus 2 (SARS-CoV-2) spike-induced syncytia formation in different cells.

**Figure supplement 6—source data 1.** Annotated, uncropped blots in *Figure 2—figure supplement 6*.

**Figure supplement 6—source data 2.** Raw, uncropped blots in *Figure 2—figure supplement 6*.

**Figure supplement 6—source data 3.** Source data of the individual points in *Figure 2—figure supplement 6*.

**Figure supplement 7.** Interleukin-1β (IL-1β) inhibits severe acute respiratory syndrome coronavirus (SARS-CoV) and MERS-CoV spike-induced cell-cell fusion.

**Figure supplement 7—source data 1.** Annotated, uncropped blots in *Figure 2—figure supplement 7*.

*Figure 2 continued on next page*

*Figure 2 continued*

**Figure supplement 7—source data 2.** Raw, uncropped blots in *Figure 2—figure supplement 7*.

**Figure supplement 7—source data 3.** Source data of the individual points in *Figure 2—figure supplement 7*.

release pathway of IL-1β from innate immune cells (*Weber et al., 2010*), we applied IL-1β for further experiments.

In order to validate the effect of IL-1β on cell-cell fusion during authentic SARS-CoV-2 infection, we pre-treated ACE2-expressing cells with IL-1β before inoculating Delta or WT authentic SARS-CoV-2. Cell lysates were used for the detection of SARS-CoV-2 spike and N protein 24 hpi (*Figure 2E*). To this end, it was found that IL-1β reduced S2′ cleavage and N protein levels compared to the control group during such infection both in HEK293T-ACE2 (*Figure 2F* and *Figure 2—figure supplement 3A*) and in Caco-2 cells (*Figure 2G*). Meanwhile, IL-1β inhibited authentic SARS-CoV-2-induced syncytia formation (*Figure 2H and I* and *Figure 2—figure supplement 3B and C*). Thus, these results verified that IL-1β inhibits authentic SARS-CoV-2-induced cell-cell fusion in various target cells.

As expected, innate immune cells activated by TLR ligands secreted IL-1β into the cell culture supernatants (*Figure 2—figure supplement 4A and B*). We employed IL-1 receptor antagonist (IL-1RA) to block IL-1 receptor on target cells, and found that IL-1RA treatment reduced the inhibitory effect of PGN-stimulated-THP-1 cell culture supernatant on cell-cell fusion (*Figure 2J* and *Figure 2—figure supplement 4C*). With another note, TLR2 was essential for THP-1 cells to release IL-1β in response to TLR2 ligands (*Figure 2—figure supplement 4D*). More importantly, the cell culture supernatants of TLR2-knockout THP-1 cells stimulated by TLR2 ligands had no effect on the bioluminescence signal, while the cell culture supernatants from WT THP-1 cells stimulated by the same TLR2 Ligands significantly reduced the bioluminescence signal (*Figure 2—figure supplement 4E*). In addition, pre-treatment with TAK1 inhibitor (5Z-7) or IKKβ inhibitor (TPCA1) in WT THP-1 cells prevented IL-1β secretion after PGN stimulation (*Figure 2—figure supplement 4F*), as well as eliminated the inhibitory effect of PGN-stimulated WT THP-1 cell culture supernatant on SARS-CoV-2 spike-induced cell-cell fusion (*Figure 2—figure supplement 4G*). In parallel, pre-treatment with these inhibitors in PBMCs showed the same results (*Figure 2—figure supplement 4H and I*). These data suggested that TLR-knockout or inhibitors targeting the respective TLR signaling prevented innate immune cells from releasing IL-1β into supernatants, which led to failed inhibition of SARS-CoV-2 spike-induced cell-cell fusion. These findings thus further verify that IL-1β is an important host factor inhibiting SARS-CoV-2-induced cell-cell fusion.

To investigate the effector function of IL-1 on cells expressing SARS-CoV-2 spike (donor cells) and neighboring cells expressing ACE2 (acceptor cells), we pre-treated HEK293T-S or HEK293T-ACE2 cells or both with IL-1β, then co-cultured after washing with phosphate buffered saline (PBS); cells were then analyzed by the quantitative and qualitative models (*Figure 2—figure supplement 5A*). Notably, pre-treatment of either HEK293T-S or HEK293T-ACE2 cells with IL-1β alone reduced bioluminescence signal and S2′ cleavage; when IL-1β pre-treatment on both HEK293T-S and HEK293T-ACE2 cells was applied, bioluminescence signal and S2′ cleavage were further reduced (*Figure 2—figure supplement 5B*). Furthermore, we also applied Vero E6-overexpressing ACE2 cell line (Vero E6-ACE2) and human Calu-3 cells as acceptor cells, and found that pre-treatment of either HEK293T-S or Vero E6-ACE2 cells with IL-1β alone reduced part of S2′ cleavage, while IL-1β pre-treatment of both HEK293T-S and Vero E6-ACE2 cells led to further reduction of S2′ cleavage (*Figure 2—figure supplement 5C*), and the same results were observed in the case of Calu-3 as acceptor cells (*Figure 2—figure supplement 5D*). Accordingly, fluorescence imaging also showed that IL-1β significantly reduced the area of syncytia (*Figure 2—figure supplement 6A and B*). Notably, IL-1β reduced the bioluminescence signal and S2′ cleavage in different SARS-CoV-2 variants (*Figure 2—figure supplement 6C–E*). Therefore, these results suggest that IL-1β acts on both donor and acceptor cells to inhibit SARS-CoV-2 spike-induced cell-cell fusion in various cell lines.

Of note, SARS-CoV (*Belouzard et al., 2009*) and MERS-CoV (*Straus et al., 2020*) spike proteins also induce cell-cell fusion in target cells. Therefore, we further explored whether IL-1β was also able to inhibit SARS-CoV and MERS-CoV spike-induced cell-cell fusion in ACE2- or dipeptidyl peptidase-4 (DPP4)-expressing cells by bioluminescence assay, immunoblotting, and a modified *stop-mCherry* fluorescent model, wherein mCherry reporter is only expressed when Cre excises the Stop cassette inside the fused syncytia (*Figure 2—figure supplement 7A*). Similar to SARS-CoV-2 spike-induced

cell-cell fusion, IL-1β also reduced bioluminescence signal (*Figure 2—figure supplement 7B and C*), S2' cleavage (*Figure 2—figure supplement 7D and E*), and the area of syncytium (*Figure 2—figure supplement 7F and G*) in these cell-cell fusion systems. Thus, IL-1β possesses a broad spectrum to inhibit cell-cell fusion induced by different coronaviruses.

## IL-1β inhibits SARS-CoV-2-induced cell-cell fusion through IL-1R1/MyD88/IRAK/TRAF6 pathway

To investigate the mechanism of IL-1β inhibition on SARS-CoV-2-induced cell-cell fusion, we performed gene knockout using CRISPR-Cas9 technology, in conjunction with inhibitors targeting the IL-1 receptor pathway (*Figure 3A*). First of all, in the presence of IL-1RA, IL-1β was unable to reduce bioluminescence signal and S2' cleavage (*Figure 3B*). Next, as MyD88 is the downstream adaptor for IL-1R1, we generated MyD88 knockout HEK293T cell line, wherein IL-1β was unable to reduce bioluminescence signal (*Figure 3C*) and S2' cleavage (*Figure 3—figure supplement 1A*). In addition, we found that IL-1β was unable to reduce bioluminescence signal and S2' cleavage in the presence of IRAK1/4 inhibitor (*Figure 3D*). Furthermore, IL-1β was unable to reduce bioluminescence signal (*Figure 3E*) and S2' cleavage (*Figure 3—figure supplement 1B*) in TRAF6 knockout HEK293T cell line. These results suggested that IL-1β inhibits SARS-CoV-2 spike-induced cell-cell fusion through IL-1R1-MyD88-IRAK-TRAF6 pathway.

Intriguingly, when we tested TAK1, a downstream molecule of TRAF6 for the potential involvement in the signaling, it was found that IL-1β still reduced bioluminescence signal and S2' cleavage in TAK1 knockout (*sgMAP3K7*) HEK293T cell line (*Figure 3—figure supplement 2A*). Moreover, we found that in the presence of TPCA1, an IKKβ inhibitor, IL-1β still inhibited bioluminescence signal and S2' cleavage as well (*Figure 3—figure supplement 2B*). In addition, although IL-1β upregulated the mRNA transcription levels of NF-κB pathway-related genes, such as *RELB*, *NFKBIA*, and *NFKB1* (*Figure 3—figure supplement 2C*), IL-1β still reduced the bioluminescence signal after these NF-κB pathway-related genes knockout (*Figure 3—figure supplement 2D*). Taken together, these results demonstrated that IL-1β inhibits SARS-CoV-2 spike-induced cell-cell fusion independent from the TAK1-IKKβ-NF-κB signaling cascade.

Furthermore, we validated these findings in authentic SARS-CoV-2-infected Caco-2 and Calu-3 cells. Consist with the results from HEK293T cells, IL-1β failed to reduce S2' cleavage and N protein levels in the presence of IRAK1/4 inhibitor, whereas it still reduced S2' cleavage and N protein amounts in the presence of the IKKβ inhibitor TPCA1 in Caco-2 (*Figure 3F*) and Calu-3 cells (*Figure 3G*).

## IL-1β inhibits SARS-CoV-2-induced cell-cell fusion through RhoA/ROCK-mediated actin bundle formation at the cell-cell junction

It has been reported that IL-1β activates RhoA signaling via MyD88 and IRAK, which is a pathway independent from IKKβ (*Chen et al., 2002*). As a major downstream effector of RhoA, ROCK phosphorylates substrates that are involved in the regulation of the actin cytoskeleton, cell attachment, and cell motility (*Riento and Ridley, 2003*). Therefore, we set out to detect the active level of RhoA through pull-down assay. To this end, we verified that IL-1β activated RhoA signaling in sgControl HEK293T cells but not in sg*MyD88*- or *TRAF6*-HEK293T cells (*Figure 4A*). To directly visualize the distribution of endogenous GTP-RhoA (active RhoA), we used a location biosensor derived from the carboxy terminus of anillin (GFP-AHPH) (*Priya et al., 2015*; *Sun et al., 2015*). Interestingly, IL-1β significantly increased the fluorescence intensity of GFP-AHPH in sgControl HEK293T cells, but had no effect in sg*MyD88*- and sg*TRAF6*-HEK293T cells (*Figure 4B and C*).

To investigate whether IL-1β inhibits SARS-CoV-2 spike-induced cell-cell fusion through the RhoA/ROCK pathway, we co-transfected GFP-AHPH in ACE2-expressing cells, then co-cultured with Spike-expressing cells at different time points. In the process of syncytia formation, cell-cell contact established between S-expressing cells and ACE2-expressing cells, and GFP-AHPH localized distally from cell-cell junction in the early stage of syncytia formation. With the enlargement of syncytium, GFP-AHPH is visualized at the periphery of syncytium (*Figure 4D*, top panel, and *Figure 4—figure supplement 1A*). However, in IL-1β-treated group, GFP-AHPH foci is enriched to the cell-cell junction in the early stage. Over time, GFP-AHPH was recruited more to the cell-cell junction between S-expressing cells and ACE2-expressing cells, preventing further cell-cell fusion (*Figure 4D*, bottom panel, and

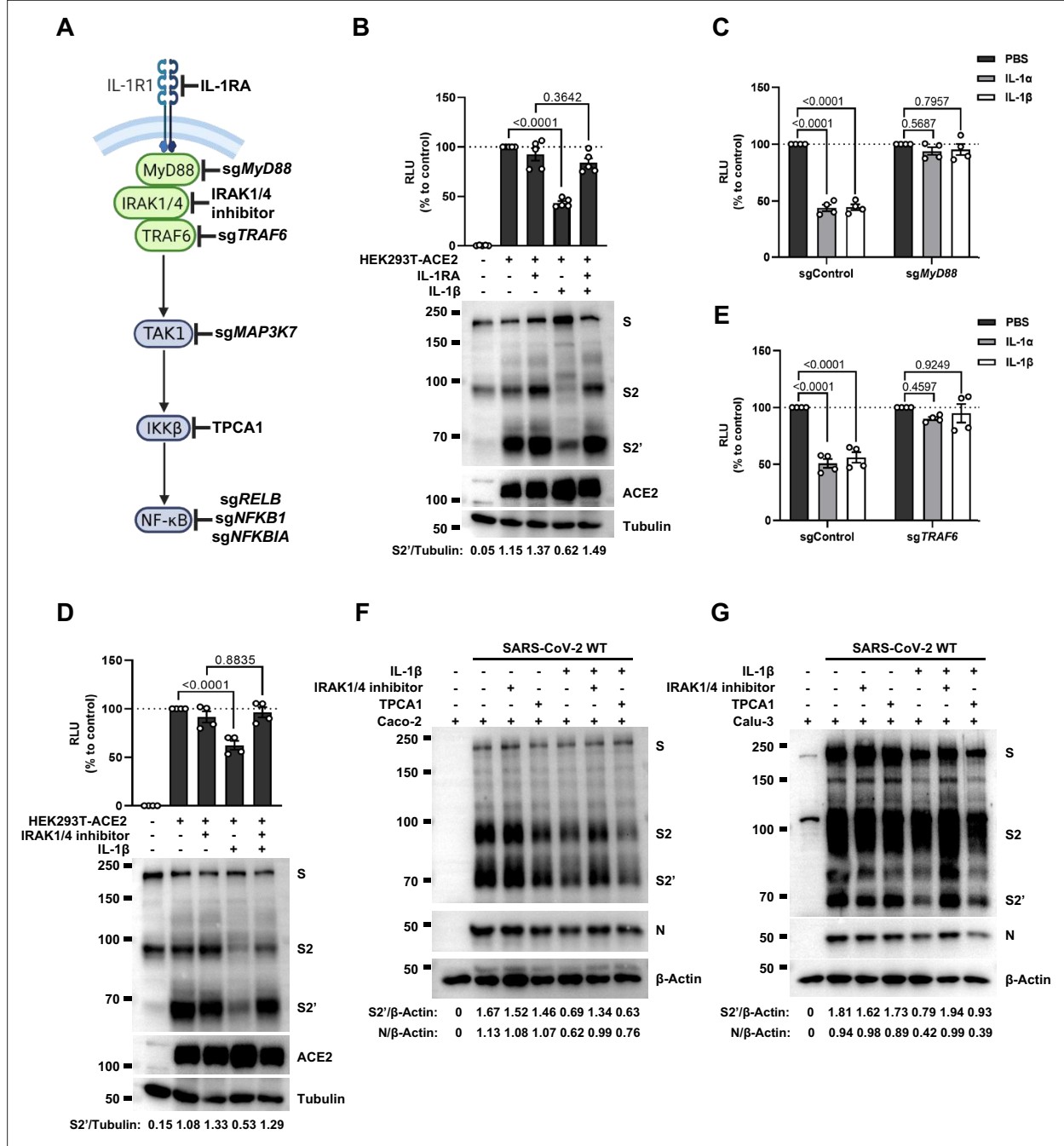

**Figure 3.** Interleukin-1β (IL-1β) inhibits severe acute respiratory syndrome coronavirus 2 (SARS-CoV-2)-induced cell-cell fusion through the IL-1R1/MyD88/IRAK/TRAF6 pathway. (**A**) Schematics of gene knockout or inhibitor treatment in the IL-1 receptor pathway. (**B**) Luciferase activity (relative luminescence units [RLU]) measured from HEK293T cell lysates and immunoblots showing full-length spike, S2, cleaved S2', and ACE2 collected from HEK293T-S and HEK293T-ACE2 pre-treated with 1000 ng/mL IL-1 receptor antagonist (IL-1RA) for 30 min, then treated with 1 ng/mL IL-1β for 16 hr. Data and blots are representative of five individual repeats. Numbers below the blots indicated the intensity of S2' versus Tubulin. (**C**) Luciferase activity (RLU) measured from cell lysates collected from 10 ng/mL IL-1α or 1 ng/mL IL-1β-treated sgControl or sg*MyD88* HEK293T cell-cell fusion system for 16 hr. Data are representative of four individual repeats and displayed as individual points with mean ± SEM. (**D**) Luciferase activity (RLU) measured from HEK293T cell lysates and immunoblots showing full-length spike, S2, cleaved S2', and ACE2 collected from HEK293T-S and HEK293T-ACE2 pre-treated with 2 μM IRAK1/4 inhibitor for 30 min, then treated with 1 ng/mL IL-1β for 16 hr. Data and blots are representative of four individual repeats. Numbers below the blots indicated the intensity of S2' versus Tubulin. (**E**) Luciferase activity (RLU) measured from cell lysates collected from 10 ng/mL IL-1α- or 1 ng/mL IL-1β-treated sgControl or sg*TRAF6* HEK293T cell-cell fusion system for 16 hr. Data are representative of four individual repeats and displayed as individual points with mean ± SEM. (**F**) Immunoblots showing full-length spike, S2, cleaved S2', and N collected from Caco-2 cells, which were pre-treated with 2 μM IRAK1/4 inhibitor and 10 ng/mL IL-1β for 1 hr, then infected with authentic SARS-CoV-2 for 24 hr. Blots are representative of three independent

*Figure 3 continued on next page*

*Figure 3 continued*

experiments. Numbers below the blots indicated the intensity of S2' or N versus β-Actin. (**G**) Immunoblots showing full-length spike, S2, cleaved S2', and N collected from Calu-3 cells, which were infected with authentic SARS-CoV-2 for 1 hr, then washed with phosphate buffered saline (PBS) before treated with 2 μM IRAK1/4 inhibitor and 10 ng/mL IL-1β for 24 hr. Blots are representative of three independent experiments. Numbers below the blots indicated the intensity of S2' or N versus β-Actin.

The online version of this article includes the following source data and figure supplement(s) for figure 3:

**Source data 1.** Annotated, uncropped blots in *Figure 3*.

**Source data 2.** Raw, uncropped blots in *Figure 3*.

**Source data 3.** Source data of the individual points in *Figure 3*.

**Figure supplement 1.** Interleukin-1β (IL-1β) inhibits severe acute respiratory syndrome coronavirus 2 (SARS-CoV-2) spike-induced cell-cell fusion through the IL-1R1/MyD88/IRAK/TRAF6 pathway.

**Figure supplement 1—source data 1.** Annotated, uncropped blots in *Figure 3—figure supplement 1*.

**Figure supplement 1—source data 2.** Raw, uncropped blots in *Figure 3—figure supplement 1*.

**Figure supplement 2.** Interleukin-1β (IL-1β) inhibits severe acute respiratory syndrome coronavirus 2 (SARS-CoV-2) spike-induced cell-cell fusion independent from the TAK1/IKKβ/NF-$\kappa$B pathway.

**Figure supplement 2—source data 1.** Annotated, uncropped blots in *Figure 3—figure supplement 2*.

**Figure supplement 2—source data 2.** Raw, uncropped blots in *Figure 3—figure supplement 2*.

**Figure supplement 2—source data 3.** Source data of the individual points in *Figure 3—figure supplement 2*.

*Figure 4—figure supplement 1B*). Cartoon schematics inserted in the imaging data illustrate such findings in a modeled manner.

It has been reported that RhoA initiates actin arc formation (*Dupraz et al., 2019*; *Stern et al., 2021*), so we further explored the changes of actin cytoskeleton during SARS-CoV-2 spike-induced cell-cell fusion. We co-transfected constitutively activated RhoA L63 (*Nobes and Hall, 1999*) (RhoA-CA) plasmid with spike or ACE2 in HEK293T cells, and found that constitutive activation of RhoA enriches actin filaments (F-actin) at cell-cell junction (*Figure 4E* and *Figure 4—figure supplement 2A*) and clearly reduces the bioluminescence signal and S2' cleavage in a dose-dependent manner (*Figure 4—figure supplement 2B*). Moreover, we observed that F-actin at cell-cell junction between S-expressing cells and ACE2-expressing cells was gradually disappeared along with cell-cell fusion in the early stages of syncytia formation. With the formation and enlargement of syncytium, F-actin of syncytium is preferably distributed peripherally (*Figure 4F*, top panel, and *Figure 4—figure supplement 2C*). However, IL-1β activated RhoA to initiate actin bundles formation at cell-cell junction, the formation of these actin bundles potentially generates barriers and prevents membrane fusion between S-expressing cells and ACE2-expressing cells. Even with the prolonged co-culture time, IL-1β-induced actin bundles formed at cell junctions consistently inhibited further syncytia formation (*Figure 4F*, middle panel, and *Figure 4—figure supplement 2D*). Of note, ROCK inhibitor Y-27632 prevents the formation of actin bundles (*van der Heijden et al., 2008*; *Watanabe et al., 1999*). Here, we found that the ROCK inhibitor Y-27632 treatment prevented the formation of IL-1β-induced actin bundles at cell-cell junctions, thus promoted membrane fusion and cytoplasmic exchange between S-expressing cells and ACE2-expressing cells and restored syncytia formation (*Figure 4F*, bottom panel, and *Figure 4—figure supplement 3A*).

Importantly, upon authentic SARS-CoV-2 infection, we observed consistent results: immunofluorescence (IF) staining showed GFP-AHPH moving to the opposite of cell-cell junction and located peripherally with syncytia formation (*Figure 5A*, top panel, and *Figure 5—figure supplement 1A*), while upon IL-1β treatment, GFP-AHPH located to the cell-cell junction of infected cells and neighboring cells (*Figure 5A*, bottom panel, and *Figure 5—figure supplement 1B*). In parallel, staining results showed that F-actin at the cell-cell junction were disassembled during authentic SARS-CoV-2 infection; with the formation of syncytium, F-actin was mainly distributed peripherally. However, actin bundles formed at cell-cell junction upon IL-1β inhibition of membrane fusion and further syncytia formation (*Figure 5B* and *Figure 5—figure supplement 1C–E*). Together, these data revealed that IL-1β induced the formation of actin bundles at the cell-cell junction of SARS-CoV-2-infected cells and neighboring cells through RhoA/ROCK pathway, which inhibited SARS-CoV-2-induced cell-cell fusion.

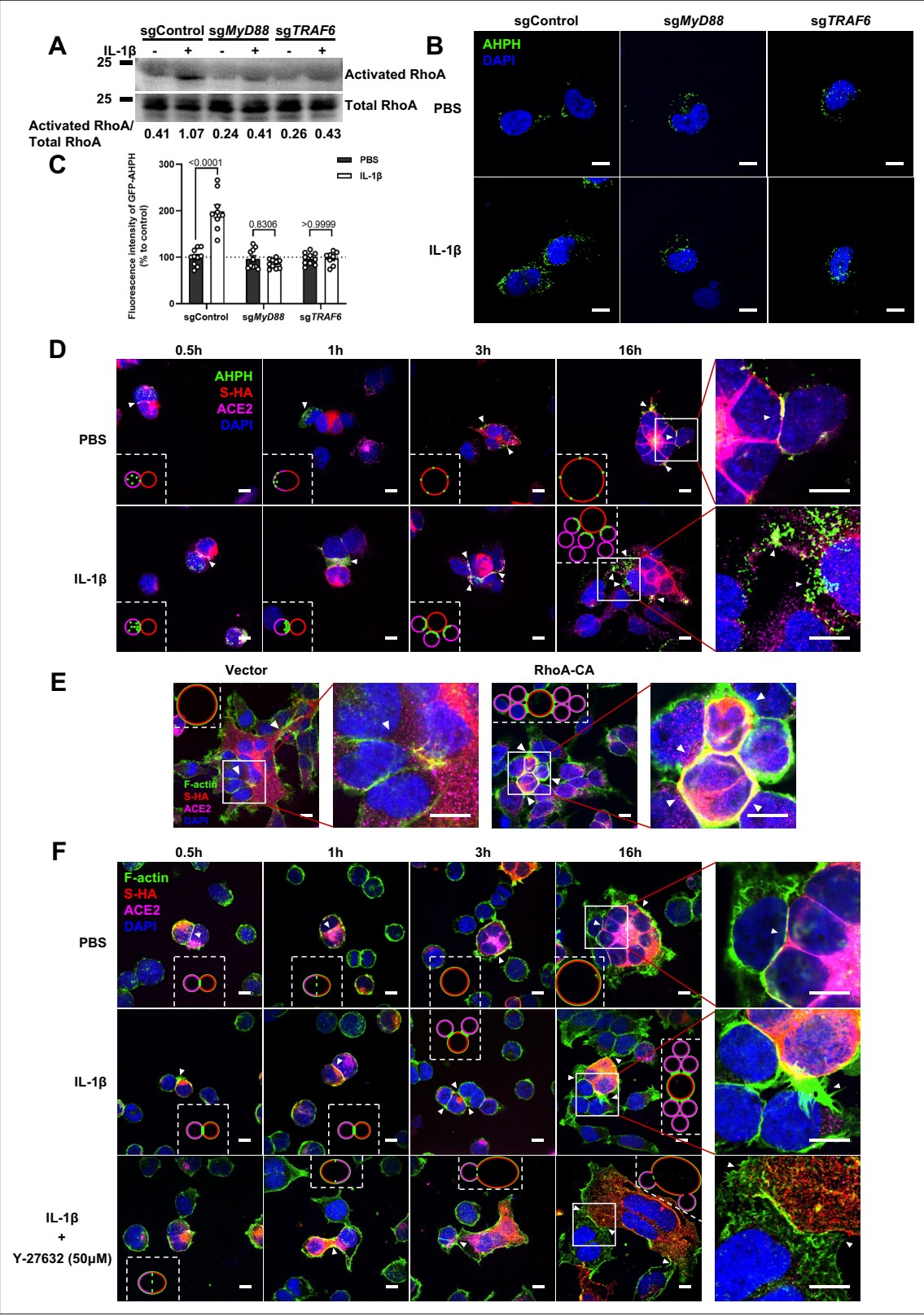

**Figure 4.** Interleukin-1β (IL-1β) inhibits severe acute respiratory syndrome coronavirus 2 (SARS-CoV-2)-induced cell-cell fusion through RhoA/ROCK-mediated actin bundles assembly at cell-cell junction. (**A**) GTP-RhoA pull-down assay to detect the active level of RhoA in sgControl, sg*MyD88,* and sg*TRAF6* HEK293T cells after 1 ng/mL IL-1β treatment for 30 min. Immunoblots showing activated RhoA and total RhoA. Blots are representative of three independent experiments. Numbers below the blots indicated the intensity of active RhoA versus total RhoA. (**B**) Representative confocal images

*Figure 4 continued on next page*

*Figure 4 continued*

of GFP-AHPH after 1 ng/mL IL-1β treatment for 30 min in sgControl, sg*MyD88,* and sg*TRAF6* HEK293T cells. Scale bars, 10 μm. (**C**) Quantification of fluorescence intensity of GFP-AHPH in (B). Data are representative of eight individual repeats. (**D**) Representative confocal images of GFP-AHPH localization with or without 1 ng/mL IL-1β treatment at different time points of syncytia formation in HEK293T-S-HA and HEK293T-ACE2 cells. Schematics with green dots in the white dashed line boxes representing GFP-AHPH, red cycles representing S-expressing cells, and magenta cycles representing ACE2-expressing cells. White arrowheads indicate the localization of GFP-AHPH, scale bars, 10 μm. Images are representative of three independent experiments. (**E**) Representative confocal images of F-actin stained with phalloidin-488 in transfected vector or 20 ng RhoA-CA HEK293T-S-HA and HEK293T-ACE2 cells. Schematics with green lines in the white dashed line boxes representing actin bundles, red cycles representing S-expressing cells, and magenta cycles representing ACE2-expressing cells. Scale bars, 10 μm. Images are representative of three independent experiments. (**F**) Representative confocal images of F-actin stained with phalloidin-488 in the presence or absence of 1 ng/mL IL-1β or 50 μM Y-27632 treatment at different time points of syncytia formation in HEK293T-S-HA and HEK293T-ACE2 cells. Schematics with green lines in the white dashed line boxes representing actin bundles, red cycles representing S-expressing cells, and magenta cycles representing ACE2-expressing cells. White arrowheads (E and F) indicate the enrichment or disappearance of F-actin, scale bars, 10 μm. Images are representative of three independent experiments.

The online version of this article includes the following source data and figure supplement(s) for figure 4:

**Source data 1.** Annotated, uncropped blots in *Figure 4*.

**Source data 2.** Raw, uncropped blots in *Figure 4*.

**Source data 3.** Source data of the individual points in *Figure 4*.

**Figure supplement 1.** Single-channel confocal images.

**Figure supplement 2.** Single-channel confocal images.

**Figure supplement 2—source data 1.** Annotated, uncropped blots in *Figure 4—figure supplement 2*.

**Figure supplement 2—source data 2.** Raw, uncropped blots in *Figure 4—figure supplement 2*.

**Figure supplement 2—source data 3.** Source data of the individual points in *Figure 4—figure supplement 2*.

**Figure supplement 3.** Single-channel confocal images.

---

To further investigate the role of RhoA/ROCK pathway in inhibiting SARS-CoV-2 induced cell-cell fusion, we found that HEK293T-ACE2 (*Figure 5C*), Caco-2 (*Figure 5—figure supplement 2A*), and Calu-3 cells (*Figure 5D*) expressing RhoA-CA clearly reduced S2' cleavage and N protein levels compared to the control group during authentic SARS-CoV-2 infection. Meanwhile, we observed that constitutive activation of RhoA enriches actin bundles at cell-cell junction, thus preventing SARS-CoV-2-induced cell-cell fusion in authentic SARS-CoV-2 infected Caco-2 (*Figure 5—figure supplement 2B and C*) and Calu-3 cells (*Figure 5E* and *Figure 5—figure supplement 2D*). In addition, we examined the potential effect of RhoA-CA on ACE2 and found that it did not affect Spike protein binding to ACE2 (*Figure 5—figure supplement 2E*), nor ACE2 distribution on the cell surface (*Figure 5—figure supplement 2F and G*). We also observed that IL-1β treatment did not change ACE2 or Spike protein distribution on the cell surface (*Figure 5—figure supplement 3A–D*).

Notably, ROCK inhibitor Y-27632 treatment increased bioluminescence signal and S2' cleavage in a dose-dependent manner, promoting syncytia formation. When treated with lower concentrations of Y-27632, IL-1β eliminated Y-27632-enhanced cell-cell fusion. However, IL-1β was unable to inhibit cell-cell fusion in the presence of higher concentrations of Y-27632 (*Figure 5—figure supplement 3E*). Furthermore, we verified that IL-1β was unable to reduce S2' cleavage and N protein levels in the presence of Y-27632 in authentic SARS-CoV-2-infected Caco-2 (*Figure 5—figure supplement 3F*), Calu-3 cells (*Figure 5F*), and primary human lung cells (*Figure 5G*). IF results also confirmed that the elimination of IL-1β induced actin bundles by Y-27632 in Caco-2 (*Figure 5—figure supplements 3G and 4A*) and Calu-3 cells (*Figure 5H* and *Figure 5—figure supplement 4B*). These results indicated that preventing the formation of RhoA/ROCK-mediated actin bundles at cell-cell junction promotes SARS-CoV-2-induced cell-cell fusion.

## IL-1β restricts SARS-CoV-2 transmission via induction of actin bundles in vivo

To demonstrate the role of IL-1β in controlling SARS-CoV-2 transmission in vivo, BALB/c mice were infected with authentic SARS-CoV-2 B.1.351 after IL-1β or IL-1RA+IL-1β pre-treatment (*Figure 6—figure supplement 1A*). Interestingly, the results of this experiment showed that in mice with IL-1β treatment, the body weight loss was less than in the PBS control group, while IL-1β was unable to improve body weight in the presence of IL-1RA (*Figure 6—figure supplement 1B*). According to

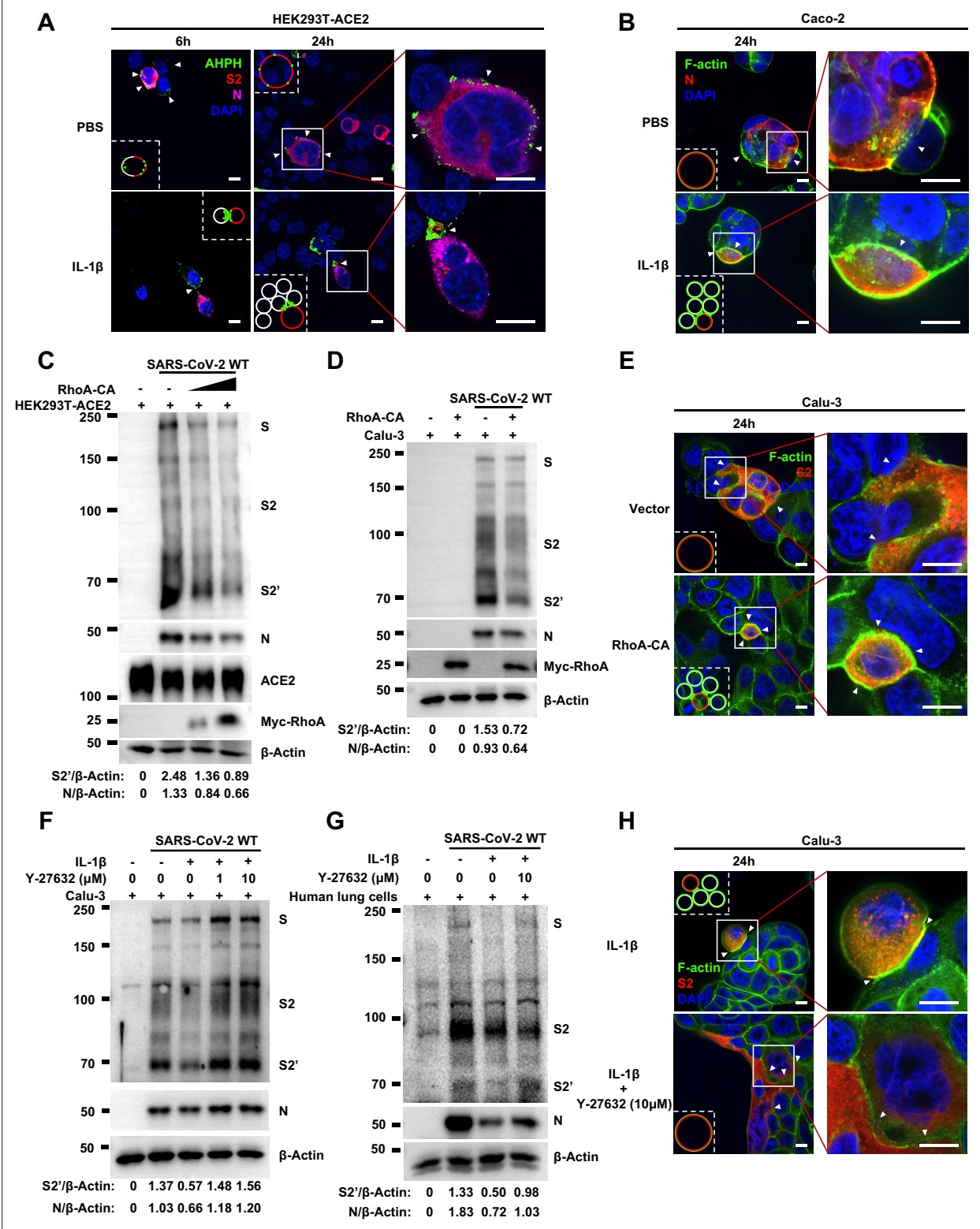

**Figure 5.** Activation of RhoA/ROCK pathway prevents authentic severe acute respiratory syndrome coronavirus 2 (SARS-CoV-2)-induced cell-cell fusion via forming actin bundles. (**A**) Representative confocal images of GFP-AHPH localization with or without 1 ng/mL interleukin-1β (IL-1β) treatment in 0.5 multiplicity of infection (MOI) wild-type (WT) authentic SARS-CoV-2-infected HEK293T-ACE2 cells at 6 and 24 hr post-infection (hpi). Schematics with green dots in the white dashed line boxes representing GFP-AHPH, red cycles representing SARS-CoV-2-infected cells, and white cycles representing neighboring cells. White arrowheads indicate the localization of GFP-AHPH, scale bars, 10 μm. Images are representative of three independent

*Figure 5 continued on next page*

*Figure 5 continued*

experiments. (**B**) Representative confocal images of F-actin stained with phalloidin-488 in the presence or absence of 1 ng/mL IL-1β treatment upon 0.5 MOI WT authentic SARS-CoV-2 infection of Caco-2 cells at 24 hpi. Schematics with green lines in the white dashed line boxes representing actin bundles, red cycles representing SARS-CoV-2-infected cells, and white cycles representing neighboring cells. Scale bars, 10 μm. Images are representative of three independent experiments. (**C**) Immunoblots of WT SARS-CoV-2 S, S2, cleaved S2′, N, and Myc-RhoA collected from HEK293T-ACE2 cells, which were transfected with vector, 10 or 20 ng RhoA-CA before infection with 0.5 MOI authentic SARS-CoV-2 WT strain for 24 hr. Blots are representative of three individual experiments. Numbers below the blots indicated the intensity of S2′ or N versus β-Actin. (**D**) Immunoblots of WT SARS-CoV-2 S, S2, cleaved S2′, N, and Myc-RhoA collected from lentivirus-transduced Calu-3 cells expressing vector or RhoA-CA, infected with WT authentic SARS-CoV-2 for 24 hr. Blots are representative of three individual experiments. Numbers below the blots indicated the intensity of S2′ or N versus β-Actin. (**E**) Representative confocal images of F-actin stained with phalloidin-488 from Calu-3 cells described in (D). Schematics with green lines in the white dashed line boxes representing actin bundles, red cycles representing S-expressing cells, scale bars, 10 μm. Images are representative of four independent experiments. (**F, G**) Immunoblots of WT SARS-CoV-2 S, S2, cleaved S2′, and N collected from Calu-3 cells (F) or primary human lung cells (G), which were infected with authentic SARS-CoV-2 for 1 hr, then washed with phosphate buffered saline (PBS) before being treated with different concentrations of Y-27632 and 10 ng/mL IL-1β for 24 hr. Blots are representative of three independent experiments. Numbers below the blots indicated the intensity of S2′ or N versus β-Actin. (**H**) Representative confocal images of F-actin stained with phalloidin-488 in Calu-3 cells described in (F). Schematics with green lines in the white dashed line boxes representing actin bundles, red cycles representing S-expressing cells. White arrowheads (B, E, and H) indicate the enrichment or disappearance of F-actin, scale bars, 10 μm. Images are representative of four independent experiments.

The online version of this article includes the following source data and figure supplement(s) for figure 5:

**Source data 1.** Annotated, uncropped blots in *Figure 5*.

**Source data 2.** Raw, uncropped blots in *Figure 5*.

**Figure supplement 1.** Single-channel confocal images.

**Figure supplement 2.** RhoA-CA does not affect Spike protein binding to ACE2 or ACE2 distribution on the cell surface.

**Figure supplement 2—source data 1.** Annotated, uncropped blots in *Figure 5—figure supplement 2*.

**Figure supplement 2—source data 2.** Raw, uncropped blots in *Figure 5—figure supplement 2*.

**Figure supplement 2—source data 3.** Source data of the individual points in *Figure 5—figure supplement 2*.

**Figure supplement 3.** Interleukin-1β (IL-1β) does not affect ACE2 and Spike distribution on the cell surface.

**Figure supplement 3—source data 1.** Annotated, uncropped blots in *Figure 5—figure supplement 3*.

**Figure supplement 3—source data 2.** Raw, uncropped blots in *Figure 5—figure supplement 3*.

**Figure supplement 3—source data 3.** Source data of the individual points in *Figure 5—figure supplement 3*.

**Figure supplement 4.** Single-channel confocal images.

hematoxylin and eosin (H&E) staining, tissue histopathology analysis demonstrated that the mice with IL-1β treatment carry less pulmonary injury compared to the PBS control and IL-1RA+IL-1β groups (*Figure 6A and B*). In addition, the expression level of SARS-CoV-2 N gene in the lung from IL-1β-treated mice was significantly lower than in the PBS control and IL-1RA+IL-1β-treated mice (*Figure 6C*). In addition, immunohistochemistry (IHC) staining showed that the infected area in the epithelial linings of lung tissue was significantly reduced by IL-1β treatment compared to the PBS control and IL-1RA+IL-1β groups (*Figure 6D and E*), indicating that IL-1β restricted the transmission of SARS-CoV-2 in the lung. Moreover, fluorescence staining showed that SARS-CoV-2-infected lung epithelial cells fused with neighboring cells, promoting viral transmission in the airway epithelial cells, while IL-1β induced the formation of actin bundles to restrict the syncytia formation and further viral transmission (*Figure 6F* and *Figure 6—figure supplement 1C and D*). In addition, we found that IL-1β-treated mice have no significant changes in body weight, nor liver and spleen weight compared to control mice (*Figure 6—figure supplement 2A–D*), indicating that this dose of IL-1β did not cause toxicity in vivo in the mice. Of note, when we isolated tissue cells from the IL-1β-treated mice and infected with authentic SARS-CoV-2, it was found that S2′ cleavage and N protein levels were strongly reduced in IL-1β-treated mice-derived lung and intestine tissue cells compared to control (*Figure 6G* and *Figure 6—figure supplement 2E*), suggesting that IL-1β may have protective effects on various tissue cells against SARS-CoV-2 infection in vivo.

To further verify the function and mechanism of IL-1β in controlling SARS-CoV-2 transmission in vivo, BALB/c mice were infected with authentic SARS-CoV-2 B.1.351 after IL-1β or ROCK inhibitor Y-27632+IL-1β pre-treatment (*Figure 7—figure supplement 1A*). Similar to IL-1RA, Y-27632 compromised the effect of IL-1β in preventing weight loss (*Figure 7—figure supplement 1B*). In addition, H&E staining showed that Y-27632 treatment aggravated lung injury in IL-1β-treated mice upon

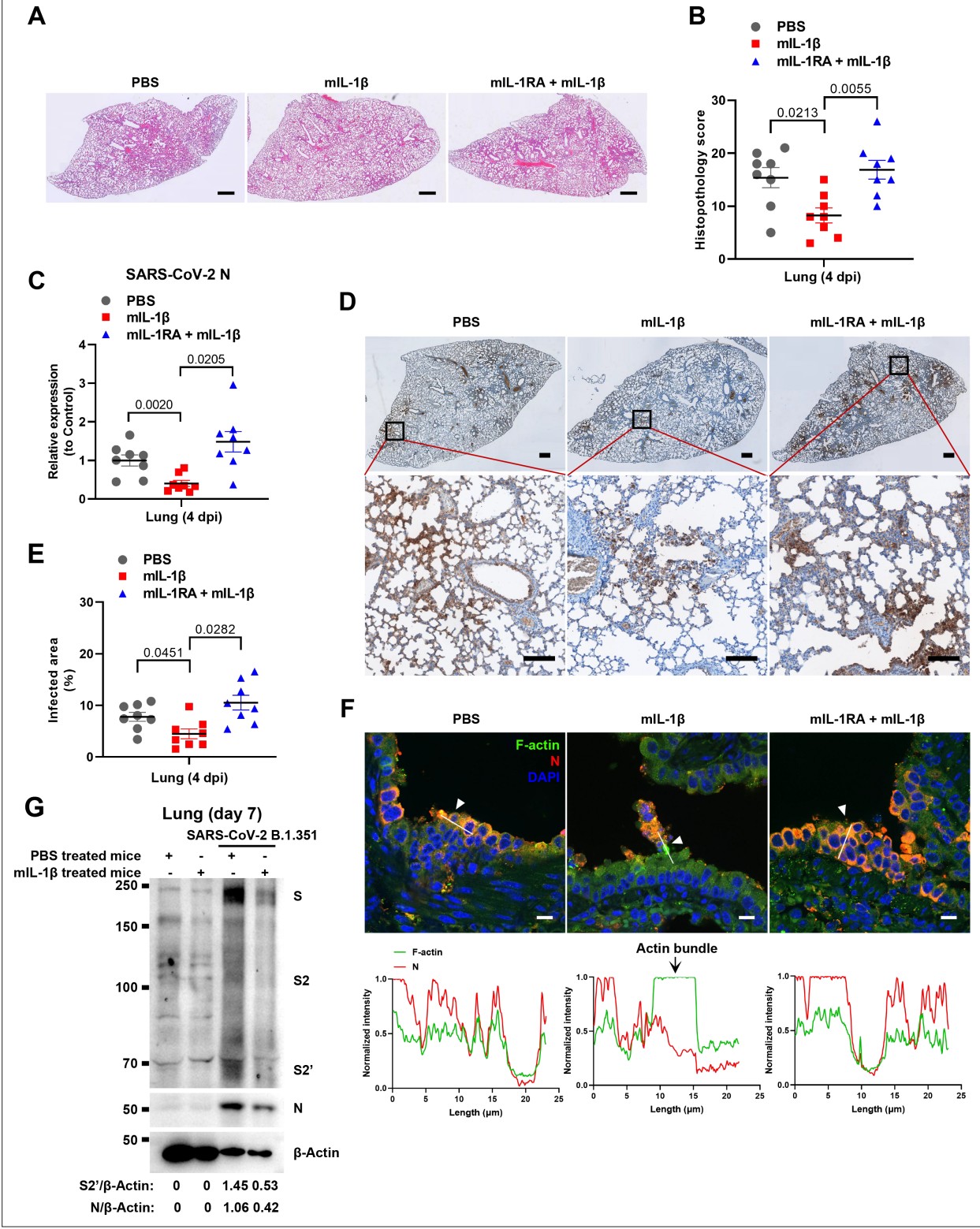

**Figure 6.** Interleukin-1β (IL-1β) restricts severe acute respiratory syndrome coronavirus 2 (SARS-CoV-2) transmission via induction of actin bundles in the lung in vivo. (**A, B**) Representative images of hematoxylin and eosin (H&E)-stained lung sections (A) and histopathology scores (B) from phosphate buffered saline (PBS); 1 μg/kg mIL-1β; 150 μg/kg mIL-1RA+mIL-1β pre-treated mice infected with SARS-CoV-2 at 4 days post-infection (dpi), scale bars are indicative of 500 μm and images are representative of eight samples. (**C**) qPCR analysis of SARS-CoV-2 N mRNA collected from infected lung tissues at 4 dpi. (**D**) Immunohistochemistry analysis of SARS-CoV-2 N staining in the lung tissue slices at 4 dpi, scale bars are indicative of 500 μm (top

*Figure 6 continued on next page*

*Figure 6 continued*

panel), 50 µm (bottom panel), and images are representative of eight samples. (**E**) The percentages of SARS-CoV-2-infected area in (D) were quantified. (**F**) Representative confocal images of F-actin stained with phalloidin-488 and SARS-CoV-2 N in the area 1 of lung tissue at 4 dpi. White arrowheads indicate syncytia formation or infected cells, scale bars are indicative of 10 µm and images are representative of three samples (top). White lines indicate SARS-CoV-2 cell-cell transmission and quantify with fluorescence intensity of F-actin and SARS-CoV-2 N (bottom). (**G**) Immunoblots of SARS-CoV-2 S, S2, cleaved S2', and N proteins collected from SARS-CoV-2 B.1.351-infected lung tissue cells, which were isolated from BALB/c mice treated with or without 1 µg/kg mIL-1β at day 7. Blots are representative of three individual mouse. Numbers below the blots indicated the intensity of S2' or N versus β-Actin.

The online version of this article includes the following source data and figure supplement(s) for figure 6:

**Source data 1.** Annotated, uncropped blots in *Figure 6*.

**Source data 2.** Raw, uncropped blots in *Figure 6*.

**Source data 3.** Source data of the individual points in *Figure 6*.

**Figure supplement 1.** Interleukin-1β (IL-1β) reduces the weight loss and viral transmission upon severe acute respiratory syndrome coronavirus 2 (SARS-CoV-2) infection in vivo.

**Figure supplement 1—source data 1.** Source data of the individual points in *Figure 6—figure supplement 1*.

**Figure supplement 2.** Treatment with 1 µg/kg mIL-1β protects mouse lung and intestinal tissue cells against severe acute respiratory syndrome coronavirus 2 (SARS-CoV-2) infection without toxicity in vivo.

**Figure supplement 2—source data 1.** Annotated, uncropped blots in *Figure 6—figure supplement 2*.

**Figure supplement 2—source data 2.** Raw, uncropped blots in *Figure 6—figure supplement 2*.

**Figure supplement 2—source data 3.** Source data of the individual points in *Figure 6—figure supplement 2*.

SARS-CoV-2 infection (*Figure 7A and B*), although Y-27632+IL-1β did not cause weight loss or lung injury in uninfected mice (*Figure 7—figure supplement 1C and D*). Moreover, Y-27632 treatment increased the expression level of SARS-CoV-2 N gene (*Figure 7C*) and infected area (*Figure 7D and E*) in the lungs of IL-1β-treated mice. Importantly, Y-27632 treatment prevented the formation of IL-1β-induced actin bundles at cell-cell junctions, thus promoted syncytia formation and further viral transmission (*Figure 7F* and *Figure 7—figure supplement 2A and B*). Furthermore, we treated BALB/c mice with PBS, IL-1β, or Y-27632+IL-1β (*Figure 7—figure supplement 2C*), then isolated the lung tissue cells for authentic SARS-CoV-2 infection. Here, it was found that S2' cleavage and N protein levels were clearly reduced in IL-1β-treated mice compared to control at day 2, while Y-27632 treatment abolished the inhibitory effect of IL-1β (*Figure 7—figure supplement 2D*). Of note, the lung tissue cells in IL-1β-treated mice remained resistant to SARS-CoV-2 infection at day 7, while the protective effect of IL-1β was abolished by Y-27632 treatment (*Figure 7G*). Taken together, IL-1β prevents the transmission of SARS-CoV-2 through inducing the formation of actin bundles via the RhoA/ROCK pathway in vivo.

## Discussion

In the present study, we explored the function of innate immune factors against SARS-CoV-2 infection. Notably, IL-1β inhibited various SARS-CoV-2 variants and other beta-coronaviruses spike-induced cell-cell fusion. Mechanistically, IL-1β activates and enriches RhoA to the cell-cell junction between SARS-CoV-2-infected cells and neighboring cells via the IL-1R-mediated signal to initiate actin bundle formation, preventing cell-cell fusion and viral spreading (*Figure 7—figure supplement 3*). These findings revealed a critical function for proinflammatory cytokines to control viral infection.

Elevated IL-1β levels in severe COVID-19 patients is central to innate immune response as it induces the expression of other proinflammatory cytokines (*Tahtinen et al., 2022*). In addition, IL-1α is also secreted during SARS-CoV-2 infection (*Xiao et al., 2021*). Of note, several therapeutic strategies have employed the inhibition of IL-1 signal in an attempt to treat SARS-CoV-2 infection (*Huet et al., 2020*; *Ucciferri et al., 2020*). Intriguingly, although anakinra, a recombinant human IL-1RA, improved clinical outcomes and reduced mortality in severe COVID-19 patients (*Cavalli et al., 2020*), it did not reduce mortality in mild-to-moderate COVID-19 patients, and even increased the probability of serious adverse events (*Tharaux et al., 2021*). With another note, IL-1 blockade significantly decreased the neutralizing activity of serous anti-SARS-CoV-2 antibodies in severe COVID-19 patients (*Della-Torre et al., 2021*). According to our finding that both IL-1β and IL-1α are able to inhibit

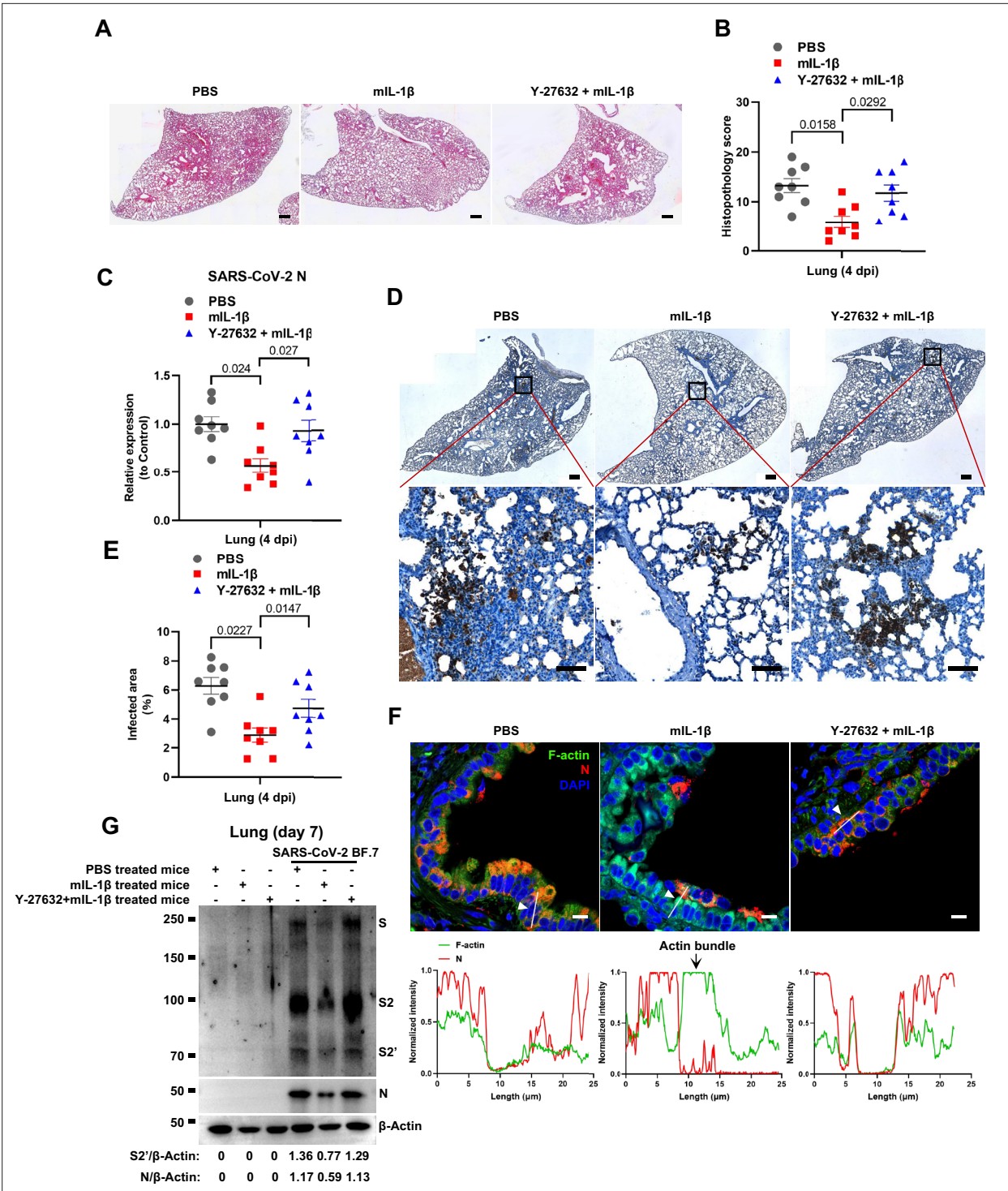

**Figure 7.** Prevention of interleukin-1β (IL-1β)-induced actin bundles by ROCK inhibitor Y-27632 promotes severe acute respiratory syndrome coronavirus 2 (SARS-CoV-2) transmission in vivo. (**A, B**) Representative images of hematoxylin and eosin (H&E)-stained lung sections (A) and histopathology scores (B) from phosphate buffered saline (PBS); 1 µg/kg mIL-1β; 1 mg/kg Y-27632+mIL-1β pre-treated mice infected with SARS-CoV-2 at 4 days post-infection (dpi), scale bars are indicative of 500 µm and images are representative of eight samples. (**C**) qPCR analysis of SARS-CoV-2 N mRNA collected from infected lung tissues at 4 dpi. (**D**) Immunohistochemistry analysis of SARS-CoV-2 N staining in the lung tissue slices at 4 dpi, scale bars are indicative of 500 µm (top panel), 50 µm (bottom panel) and images are representative of eight samples. (**E**) The percentages of SARS-CoV-2-infected area in (D) were quantified. (**F**) Representative confocal images of F-actin stained with phalloidin-488 and SARS-CoV-2 N in the area 1 of lung tissue at 4 dpi. White arrowheads indicate syncytia formation or infected cells, scale bars are indicative of 10 µm and images are representative of three samples (top). White

*Figure 7 continued*

lines indicate SARS-CoV-2 cell-cell transmission and quantify with fluorescence intensity of F-actin and SARS-CoV-2 N (bottom). (**G**) Immunoblots of SARS-CoV-2 S, S2, cleaved S2′, and N proteins collected from authentic SARS-CoV-2 BF.7-infected lung tissue cells, which were isolated from BALB/c mice treated with PBS, 1 µg/kg mIL-1β or 1 mg/kg Y-27632+1 µg/kg mIL-1β at day 7. Blots are representative of three individual mouse. Numbers below the blots indicated the intensity of S2′ or N versus β-Actin.

The online version of this article includes the following source data and figure supplement(s) for figure 7:

**Source data 1.** Annotated, uncropped blots in *Figure 7*.

**Source data 2.** Raw, uncropped blots in *Figure 7*.

**Source data 3.** Source data of the individual points in *Figure 7*.

**Figure supplement 1.** Y-27632+IL-1β do not cause weight loss and lung injury in uninfected BALB/c mice.

**Figure supplement 1—source data 1.** Source data of the individual points in *Figure 7—figure supplement 1*.

**Figure supplement 2.** Y-27632 treatment prevents the formation of interleukin-1β (IL-1β)-induced actin bundles at cell-cell junctions in vivo.

**Figure supplement 2—source data 1.** Annotated, uncropped blots in *Figure 7—figure supplement 2*.

**Figure supplement 2—source data 2.** Raw, uncropped blots in *Figure 7—figure supplement 2*.

**Figure supplement 2—source data 3.** Source data of the individual points in *Figure 7—figure supplement 2*.

**Figure supplement 3.** Graphical abstract.

---

SARS-CoV-2-induced cell-cell fusion, inhibition of IL-1 signaling may have abolished the antiviral function of IL-1, thus failing to restrict virus-induced syncytia formation and transmission.

Notably, IL-1β plays a key role in triggering vaccine-induced innate immunity, suggesting that innate immune responses play important roles in the antiviral defense by enhancing the protective efficacy of vaccines (*Eisenbarth et al., 2008*; *Tahtinen et al., 2022*). In addition, vaccination with Bacillus Calmette-Guérin (BCG) has been reported to confer nonspecific protection against heterologous pathogens, including protection against SARS-CoV-2 infection in humans and mice (*Hilligan et al., 2022*; *Lee et al., 2024*; *Rivas et al., 2021*). Moreover, lipid nanoparticle in mRNA vaccine (*Han et al., 2023*) and penton base in adenovirus vaccine (*Di Paolo et al., 2009*) can both activate innate immune cells to amplify the protective effect of vaccines, which may also be attributed to IL-1β-mediated inhibition of SARS-CoV-2-induced cell-cell fusion on top of adaptive immune responses induced by the vaccines.

With another note, patients with inherited MyD88 or IRAK4 deficiency have been reported to be selectively vulnerable to COVID-19 pneumonia. It was found that these patients' susceptibility to SARS-CoV-2 can be attributed to impaired type I IFN production, which do not sense the virus correctly in the absence of MyD88 or IRAK4 (*García-García et al., 2023*). In our study, MyD88 or IRAK4 deficiency abolished the inhibitory effect of IL-1β on SARS-CoV-2-induced cell-cell fusion, suggesting that these innate immune molecules are critical to contain SARS-CoV-2 infection, and this may be another mechanism accounting for the disease of those patients. Moreover, MyD88 signaling was essential for BCG-induced innate and type 1 helper T cell (TH1 cell) responses and protection against SARS-CoV-2, which is consistent with our fundings.

Of note, cell-cell fusion is not limited to the process of viral infection, both normal and cancerous cells can utilize this physiological process in tissue regeneration or tumor evolution (*Delespaul et al., 2020*; *Powell et al., 2011*). For example, myoblast fusion is the key process of skeletal muscle terminal differentiation, inactivation of RhoA/ROCK signaling is crucial for myoblast fusion (*Nishiyama et al., 2004*). Our current work revealed that inhibition of RhoA/ROCK signaling promoted virus-induced cell-cell fusion, possibly due to the virus hijacking of such biological process. In turn, activated RhoA/ROCK signaling inhibits virus-induced cell-cell fusion, so it can be targeted for future therapeutic development to control viral transmission. Cell-cell fusion is mediated by actin cytoskeletal rearrangements, the dissolution of F-actin focus is essential for cell-cell fusion; in contrast, syncytia formation cannot proceed if disassembly of actin filaments or bundles is prevented (*Doherty et al., 2011*; *Rodríguez-Pérez et al., 2021*). We uncovered that preventing actin bundles dissolution inhibited virus-induced cell-cell fusion, and IL-1β-induced RhoA/ROCK signal promotes actin bundle formation at cell-cell junctions. As RhoA is ubiquitously expressed by all cell types, it is currently unclear whether IL-1-mediated RhoA activation is specific toward viral infection-associated cytoskeleton modification,

or may regulate other RhoA-related processes, which is a limitation of the current work and remains to be investigated in future.

In summary, this study demonstrated the function and mechanism of IL-1β in inhibiting SARS-CoV-2-induced syncytia formation, and highlighted the function of innate immune factors including cytokines against coronaviruses transmission, thus provide potential therapeutic targets for viral control.

# Materials and methods

## Key resources table

| Reagent type (species) or resource | Designation | Source or reference | Identifiers | Additional information |
|---|---|---|---|---|
| Antibody | Rabbit polyclonal antibody (pAb) to SARS-CoV-2 S2 | Sino Biological | Cat#:40590-T62, RRID:AB_3073714 | WB (1:2000), IF (1:200) |
| Antibody | Mouse monoclonal antibody (mAb) to SARS-CoV-2 Nucleocapsid | Sino Biological | Cat#:40143-MM05, RRID:AB_2827977 | WB (1:1000), IF (1:200) |
| Antibody | Rabbit polyclonal antibody (pAb) to ACE2 | Proteintech | Cat#:21115-1-AP, RRID:AB_10732845 | WB (1:2000), IF (1:200) |
| Antibody | Rabbit polyclonal antibody (pAb) to MERS-CoV S2 antibody | Sino Biological | Cat#:40070-T62 | WB (1:1000) |
| Antibody | Rabbit monoclonal antibody (mAb) to MyD88 | Cell Signaling Technology | Cat#:4283, RRID:AB_10547882 | WB (1:1000) |
| Antibody | Rabbit monoclonal antibody (mAb) to TRAF6 | Abcam | Cat#:ab33915, RRID:AB_778572 | WB (1:1000) |
| Antibody | Rabbit polyclonal antibody (pAb) to TAK1 | Cell Signaling Technology | Cat#:4505, RRID:AB_490858 | WB (1:1000) |
| Antibody | Mouse monoclonal antibody (mAb) to Myc-Tag | Abclonal | Cat#:AE010, RRID:AB_2770408 | WB (1:2000) |
| Antibody | Mouse monoclonal antibody (mAb) to HA-Tag | Abclonal | Cat#:AE008, RRID:AB_2770404 | IF (1:200) |
| Antibody | Mouse monoclonal antibody (mAb) to HRP-conjugated β-tubulin | Abclonal | Cat#:AC030, RRID:AB_2769870 | WB (1:5000) |
| Antibody | Mouse monoclonal antibody (mAb) to β-actin | Proteintech | Cat#:66009-1-Ig, RRID:AB_2687938 | WB (1:5000) |
| Antibody | Goat anti-Mouse IgG (H+L) Highly Cross-Adsorbed Secondary Antibody, Alexa Fluor 555 | Invitrogen | Cat#:A-21424, RRID:AB_141780 | IF (1:400) |
| Antibody | Goat anti-Mouse IgG (H+L) Highly Cross-Adsorbed Secondary Antibody, Alexa Fluor 647 | Invitrogen | Cat#:A-21236, RRID:AB_2535805 | IF (1:400) |
| Chemical compound | Actin-Tracker Green-488 | Beyotime | C2201S | IF (1:100) |
| Chemical compound | DAPI | Abcam | ab228549 | IF (1:2000) |
| Chemical compound | Antifade mounting medium | Vectorlabs | H-1400-10 | N/A |
| Chemical compound | Purified LTA from *S. aureus* | Invitrogen | tlrl-pslta | 10 µg/mL |
| Chemical compound | Pam3CSK4 | Invitrogen | tlrl-pms | 1 µg/mL |
| Chemical compound | Peptidoglycan from *S. aureus* | Sigma-Aldrich | 77140 | 2 µg/mL |
| Chemical compound | LPS | Invitrogen | tlrl-eklps | 1 µg/mL |
| Chemical compound, drug | TPCA1 | Selleck | S2824 | 0.5 µM, 1 µM |
| Chemical compound, drug | 5Z-7-Oxozeaenol | Sigma-Aldrich | O9890 | 1 µM |
| Chemical compound, drug | IRAK1/4 inhibitor | Selleck | S6598 | 2 µM |
| Chemical compound, drug | Y-27632 | Selleck | S6390 | In vitro: 1 µM, 10 µM, 50 µM; In vivo: 1 mg/kg |
| Recombinant protein | Recombinant human IL-1α | Peprotech | 200-01A | 10 ng/mL |
| Recombinant protein | Recombinant human IL-1β | Peprotech | 200-01B | 1 ng/mL |
| Recombinant protein | Recombinant mouse IL-1β | Peprotech | 211-11B | In vivo: 1 µg/kg |

*Continued on next page*

*Continued*

| Reagent type (species) or resource | Designation | Source or reference | Identifiers | Additional information |
|---|---|---|---|---|
| Recombinant protein | Recombinant human IL-1RA | Peprotech | 200-01RA | 1000 ng/mL |
| Recombinant protein | Recombinant human IL-6 | Peprotech | 200-06 | 100 ng/mL |
| Recombinant protein | Recombinant human IL-8 | Peprotech | 200-08M | 100 ng/mL |
| Recombinant protein | Recombinant mouse IL-1RA | BioLegend | 769706 | In vivo: 150 µg/kg |
| Commercial assay or kit | Human IL-1β ELISA kit | R&D Systems | DY201 | N/A |
| Commercial assay or kit | RhoA pull-down activation assay Biochem kit | Cytoskeleton | BK036-S | N/A |
| Recombinant DNA reagent | pVAX1 SARS-CoV-2 spike (Wild type) plasmid | This paper | GenBank: QHD43419.1 | *Homo sapiens* codon-optimized, HA-tag at the C-terminal |
| Recombinant DNA reagent | pVAX1 SARS-CoV-2 spike (Alpha) plasmid | This paper | N/A | Truncated 19 amino acids at the C-terminal |
| Recombinant DNA reagent | pVAX1 SARS-CoV-2 spike (Beta) plasmid | This paper | N/A | Truncated 19 amino acids at the C-terminal |
| Recombinant DNA reagent | pVAX1 SARS-CoV-2 spike (Delta) plasmid | This paper | N/A | Truncated 19 amino acids at the C-terminal |
| Recombinant DNA reagent | pVAX1 SARS-CoV-2 spike (Omicron) plasmid | GeneScript | N/A | Truncated 19 amino acids at the C-terminal |
| Recombinant DNA reagent | pcDNA4.0 human ACE2 plasmid | This paper | N/A | V5-tag at the C-terminal |
| Recombinant DNA reagent | GFP-AHPH plasmid | Addgene | Cat#:71368, RRID:Addgene_71368 | N/A |
| Recombinant DNA reagent | pRK5myc RhoA L63 plasmid | Addgene | Cat#:15900, RRID:Addgene_15900 | N/A |
| Sequence-based reagent | PCR primers | This paper | PCR primers | See *Supplementary file 1* for primers used in this study |
| Sequence-based reagent | sgRNA primers | This paper | sgRNA primers | See *Supplementary file 2* for sgRNA primers used in this study |

## Cell culture and stimulation

HEK293T cells (4201HUM-CCTCC00187) were purchased from the National Science & Technology Infrastructure (NSTI) cell bank (https://www.cellbank.org.cn/). Human colon epithelial carcinoma cell line Caco-2 (catalog no. SCSP-5027) cells were obtained from Cell Bank/Stem Cell Bank, Chinese Academy of Sciences. Human lung cancer cell line Calu-3 and Vero E6-ACE2 cells were gifted from Prof. Dimitri Lavillette (Applied Molecular Virology Laboratory, Discovery Biology Department, Institut Pasteur Korea). Human monocytic cell line THP-1 (TIB-202; ATCC) was authenticated at Genetic Testing Biotechnology Corporation (Suzhou, China) using short tandem repeat analysis as described in 2012 in ANSI Standard (ASN-0002) by the ATCC Standards Development Organization. Their identity has been authenticated by the supplier and regular mycoplasma checks were performed. HEK293T and Vero E6-ACE2 cells were cultured in Gibco Dulbecco's Modified Eagle Medium (DMEM) (GE Healthcare) supplemented with 10% fetal bovine serum (FBS) (Sigma) and 1% penicillin/streptomycin (P/S) (Life Technologies) at 37°C with 5% $CO_2$ in a humidified incubator. Caco-2 and Calu-3 cells were cultured in Minimum Essential Medium supplemented with 10% FBS, 1% non-essential amino acids, and 1% P/S at 37°C with 5% $CO_2$ in a humidified incubator. THP-1 cells were cultured in Roswell Park Memorial Institute (RPMI) 1640 supplemented with 10% FBS, 1% P/S, and 50 µM 2-ME at 37°C with 5% $CO_2$ in a humidified incubator. All cells were routinely tested for mycoplasma contamination; passages between 4th and 25th were used. Human PBMCs were isolated from the peripheral blood of healthy doners (Shanghai Blood Center). This study was performed in accordance with the International Ethical Guidelines for Biomedical Research Involving Human Subjects and the principles expressed in the Declaration of Helsinki. Briefly, fresh human PBMCs were separated using Ficoll-Paque PLUS reagent (cytiva, 17144003) at 1200×*g* for 10 min at room temperature with SepMateTM-50 (SepMate, 86450). PBMCs were washed three times with filtered PBS containing 0.5% bovine

serum albumin (BSA) and 2 mM EDTA. PBMCs were counted and resuspended in RPMI 1640 medium supplemented with 1% FBS and 1% P/S.

For stimulation, THP-1 cells were seeded at $2\times10^6$ cells/mL in FBS-free RPMI 1640 and PBMCs were seeded at $1\times10^7$ cells/mL in 1% FBS RPMI 1640, then stimulated with LTA (10 µg/mL), Pam3CSK4 (1 µg/mL), PGN (2 µg/mL), LPS (1 µg/mL) for 24 hr, cell culture supernatants were collected after centrifugation at $2000\times g$ for 5 min for subsequent experiments.

## Transient transfection and cell-cell fusion assays

For transient transfections, HEK293T cells were seeded in 24-well plates at $0.5\times10^6$ cells/mL overnight. 250 ng plasmids encoding SARS-CoV-2 spike mutants or ACE2 variants were packaged in Lipofectamine 2000 (Life Technologies) and transfected for 24 hr. For luciferase assays, Spike-mediated membrane fusion, a *Cre-loxp* Firefly luciferase (*Stop-Luc*) co-expression system was introduced to enable the detection of DNA recombination events during cell-cell fusion. 200 ng Cre plasmids were co-transfected into HEK293T-S cells and 200 ng *Stop-Luc* plasmid were co-transfected into HEK293T±ACE2 cells, respectively. For visualization of syncytia formation, 100 ng ZsGreen plasmid was co-transfected with spike variants. HEK293T cells in the 24-well plates were then detached using ice-cold calcium-free PBS in the absence of trypsin and centrifuged at $600\times g$ for 4 min.

For cell-cell fusion assays, cell pellets were resuspended into complete DMEM and mixed with control HEK293T cells, or HEK293T-ACE2, Vero E6-ACE2, or Calu-3 cells at 1:1 ratio before adhesion to the 48-well or 96-well plates, cell mixes were incubated for 16 hr at 37°C. Quantification of cell-cell fusion was performed by measuring luciferase expression as relative luminescence units (RLU) 1 min by mixing cell lysates with the Bright-Glo luciferase substrate (E2610, Promega) on a Synergy H1 plate reader (Biotek). Fluorescent images showing syncytia formation were captured at endpoint using a 10× objective and 12-bit monochrome CMOS camera installed on the IX73 inverted microscope (Olympus). Attached cells and syncytia were lysed in an NP40 lysis buffer containing 0.5% (vol/vol) NP40, 25 mM Tris pH 7.3, 150 mM NaCl, 5% glycerol, and 1× EDTA-free protease inhibitor cocktail (PIC) (Roche).

## Immunoblotting

Tissue culture plates containing adherent syncytia and cell mixes were directly lysed on ice in 2× reducing Laemmli loading buffer before boiled at 95°C for 5 min. Protein samples were separated by standard Tris-glycine SDS-PAGE on 7.5% or 9.5% Tris-glycine polyacrylamide gels. Proteins were then transferred onto 0.45 µm PVDF membranes (Millipore) for wet transfer using Towbin transfer buffer. All membranes were blocked in PBS supplemented with 0.1% Tween 20 (PBST) and 2.5% BSA or 5% non-fat dry milk, before overnight incubation in primary antibodies at 4°C. Blots were labeled with HRP-tagged secondary antibodies (Jackson ImmnuoResearch) and visualized with PicoLight substrate enhanced chemiluminescence solution (Epizyme Scientific). Immunoblot images were captured digitally using a 5200 chemiluminescent imaging system (Tanon) with molecular weight markers indicated.

## Real-time PCR

$0.5\times10^6$ cells/mL HEK293T cells were seeded in 24-well plates overnight. After the cells were about 80% covered, specified stimulant was added. Upon harvesting, cells were washed with PBS for three times, and 1 mL TRIzol Reagent (15596018; Thermo Fisher Scientific) was added for full lysis at room temperature for 5 min. 250 µL chloroform was added, fully mixed at room temperature for 5 min, centrifuged at 10,000 r/min, 4°C for 10 min. After carefully removing the aqueous phase using a pipette into another 1.5 mL Eppendorf tube, some of the aqueous phase (about 1 mm above DNA layer to prevent DNA contamination) was remained. 550 µL isopropanol was added in the aqueous phase and mixed gently, then placed at –20°C for 30 min. The tubes were centrifuged at 14,000 r/min, 4°C for 20 min, and washed with 75% ethanol twice before dissolved in 30 µL DEPC water. RNA was reverse-transcribed to cDNA using a GoSript Reverse Transcription Kit (Promega). Real-time PCR was performed using SYBR Green Realtime PCR Master Mix (TOYOBO) on ABI QuantStudio 6 flex Real-time PCR System (Thermo Fisher Scientific). The RT-qPCR primer sequences for targeting genes are displayed in *Supplementary file 1*. Target genes' relative quantification was normalized to *GAPDH* as relative unit (RU).

## CRISPR/Cas9-mediated gene targeting

Gene-deficient THP-1 or HEK293T cells were generated using CRISPR/Cas9-mediated gene targeting technology. Briefly, LentiCRISPR v2 (52961; Addgene) containing sgRNA specifically targeting indicated genes were constructed. The sgRNA sequences for targeting respective genes are displayed in *Supplementary file 2*. The lentiviral particles were produced in HEK293T cells by transfection with LentiCRISPR v2-sg gene, psPAX2, VSV-G at 2:1.5:1 ratio using Lipofectamine 2000. The lentiviral particles were employed to infect THP-1 or HEK293T cells. One day post-infection (dpi), the cells were subjected to puromycin selection at a concentration of 2 µg/mL for 72 hr. Survived cells were subjected to limiting dilution in 96-well plates to obtain single clones stably knocking-out respective genes.

## RhoA pull-down assay

RhoA pull-down activation assay Biochem kit was applied for this experiment. In brief, after 1 ng/mL IL-1β treatment for 30 min, HEK293T cells were placed on ice and the culture media was aspirated off before washing cells with ice-cold PBS, then washed cells were transferred into 1.5 mL Eppendorf tubes followed with a centrifugation 600×*g*, 4°C for 5 min. Then, the cell lysis buffer with PIC was added. The tubes were immediately centrifuged at 10,000×*g*, 4°C for 1 min, then 20 µL of the lysate was saved for total RhoA, and the remaining lysate was used for pull-down assay. For pull-down assay, 10 µL rhotekin-RBD beads were mixed with 600 µg total protein, then the tubes were incubated at 4°C on a rotator for 1 hr before centrifuged at 5000×*g*, 4°C for 1 min. Next, 90% of the supernatants were carefully removed before washing beads with 500 µL wash buffer. Then, the tubes were centrifuged at 5000×*g*, 4°C for 3 min, and supernatant was carefully removed before adding 20 µL of 2× Laemmli sample buffer, then the beads were thoroughly resuspended and boiled for 2 min and analyzed through immunoblotting.

## Immunostaining and confocal microscopy

HEK293T-ACE2 cells were seeded onto sterilized poly-D-lysine (100 µg/mL) (Beyotime, ST508) treated 12 mm coverslips (Fisher Scientific, 1254580) in 24-well plates. After co-culture with HEK293T-S, cells were washed with PBS once before fixing with 4% (wt/vol) paraformaldehyde (PFA) for 20 min. Then, cells were washed twice with PBS and permeabilized with 0.1% Triton at room temperature for 10 min (for wheat germ agglutinin [WGA] staining, cells were not treated with Triton). Next, cells were washed twice with PBS and blocked with Immunol Staining Blocking Buffer (Beyotime, P0102) at room temperature for 1 hr. Primary antibodies were incubated at room temperature for 1 hr. Coverslips were then washed twice with PBS before incubation with Actin-Tracker Green-488 or secondary antibodies for 1 hr at room temperature. Coverslips were washed twice with PBS before DAPI staining for 10 min or being mounted in antifade mounting medium. Fluorescent images covering various areas on the coverslips were captured at 12-bit depth in monochrome using a 100× oil immersion objective on the Olympus SpinSR10 confocal microscope and subsequently processed using ImageJ software (NIH) with scale bars labeled.

## Authentic SARS-CoV-2 infection of cells

All experiments involving authentic SARS-CoV-2 virus in vitro were conducted in the biosafety level 3 laboratory of the Shanghai Municipal Center for Disease Control and Prevention (CDC). The experiments and protocols in this study were approved by the Ethical Review Committee of the Shanghai CDC (Permit Number: 2022-51). Briefly, HEK293T-ACE2 or Caco-2 cells were seeded into 24-well or 96-well plates at a density of $4×10^5$ cells/mL overnight, then pre-treated with different reagents for 1 hr before infection with 0.5 multiplicity of infection (MOI) Delta or WT authentic SARS-CoV-2 (B.1.617.2 and WT) for 24 hr. Calu-3 cells were seeded into 24-well or 96-well plates at a density of $4×10^5$ cells/mL overnight, infected with 0.5 MOI WT authentic SARS-CoV-2 for 1 hr, then washed with PBS before treating with different reagents for 24 hr. Bright-field images were captured to indicate the syncytia formation, cell lysates were collected for spike S2′ cleavage and N protein immunoblots.

For primary mouse tissue cells, specific pathogen-free 6-week-old female BALB/c mice were lightly anesthetized with isoflurane and intranasal treated with PBS, mIL-1β (1 µg/kg) or Y-27632 (1 mg/kg)+mIL-1β (1 µg/kg) at day 0, then mice were intraperitoneally injected with PBS, mIL-1β (1 µg/kg) or Y-27632 (1 mg/kg)+mIL-1β (1 µg/kg) at days 1 and 2. At day 7, mice were anesthetized by

intraperitoneal injection of Avertin (2,2,2-tribromoethanol, Sigma-Aldrich), the thoracic cavity and abdominal cavity were opened, an outlet was cut in the left ventricle of the mice, and then the right ventricle was perfused with PBS through the pulmonary artery to remove blood cells in the lung. Next, the lung digestive solution with HBSS 1 mL, 1 mg/mL collagenase IA, DNaseI (200 mg/mL; Roche), DispaseII (4 U/mL; Gibco), and 5% FBS was injected into the lung cavity, then the lungs were peeled off and digested for 30 min with shaking at 37°C. For the intestinal cell isolation, the intestines are peeled off and put into the intestine digestive solution containing DMEM 10 mL, 1 μM DTT, 0.25 μM EDTA, and 5% FBS, then digested for 30 min with shaking at 37°C. After digestion, the lung and intestinal cells were resuspended in DMEM supplemented with 10% FBS and 1% P/S, subsequently, the cells were infected with 1 MOI authentic SARS-CoV-2 B.1.351 or BF.7 for 24 hr. All procedures were conducted in compliance with a protocol approved by the Institutional Animal Care and Use Committee (IACUC) at Shanghai Institute of Immunity and Infection, Chinese Academy of Sciences.

For primary human lung cells, the human lung tissues were cut into small pieces of about 2 mm$^3$ and washed three times with HBSS solution containing 1% PS, digested with collagenase type I (100 mg+50 mL PBS) in an incubator at 37°C for 4 hr. Then filtered through a 70 μm filter and centrifuged at 500×$g$ for 5 min at room temperature. Lysed with 3 mL Red Blood Cell Lysis Buffer for 5 min at room temperature, and then centrifuged at 500×$g$ for 5 min at room temperature, washed twice with HBSS solution containing 1% PS. Human lung cells were resuspended in B-ALI Growth Media and seeded into 96-well plates at a density of 4×10$^5$ cells/mL overnight, infected with 0.5 MOI WT authentic SARS-CoV-2 for 1 hr, then washed with PBS before treating cells with different reagents for 24 hr. The experiments and protocols were approved by the Ethical Review Committee of the Shanghai CDC.

## Authentic SARS-CoV-2 infection of BALB/c mice

Specific pathogen-free 6-week-old female BALB/c mice were lightly anesthetized with isoflurane and intranasal treated with PBS, mIL-1β (1 μg/kg), mIL-1RA (150 μg/kg)+mIL-1β (1 μg/kg); or PBS, mIL-1β (1 μg/kg), Y-27632 (1 mg/kg)+mIL-1β (1 μg/kg) for 1 hr, then intranasally challenged with 5×10$^4$ FFU of SARS-CoV-2 B.1.351. For booster injection, mice were intraperitoneally injected with PBS, mIL-1β (1 μg/kg), mIL-1RA (150 μg/kg)+mIL-1β (1 μg/kg); or PBS, mIL-1β (1 μg/kg), Y-27632 (1 mg/kg)+mIL-1β (1 μg/kg) at 1 and 2 dpi. Mice were monitored daily for weight loss. Lungs were removed into TRIzol or 4% PFA at 4 dpi. Animal experiments related to authentic SARS-CoV-2 were conducted in Guangzhou Customs District Technology Center BSL-3 Laboratory (Permit Number: IQTC202302).

## Pulmonary histopathology

Lungs were collected from mice infected with SARS-CoV-2 at 4 dpi and fixed in 4% PFA (Bioss) for 12 hr followed by dehydrating, embedded in paraffin for sectioning, then stained with H&E, IHC, or IF. H&E and IHC data were analyzed by PerkinElmer Vectra 3, IF results were analyzed by Olympus SpinSR10 confocal microscope. The pathological scores were judged according to previous work (*Curtis et al., 1990*).

## Statistics analysis

Bar graphs were presented as mean values ± standard error of mean (SEM) with individual data points. All statistical analyses were carried out with the Prism software v8.0.2 (GraphPad). Data with multiple groups were analyzed using matched one-way ANOVA followed by Sidak's post hoc comparisons. Statistical significance p values were indicated between compared groups and shown on figures.

## Acknowledgements

We thank Qiuhong Guo, Prof. Dimitri Lavillette, and Prof. Gary Wong for their experimental supports and reagents used in this work. This study is supported by grants from Natural Science Foundation of China (92269202, 82825001, 92054104, 82402593), National Key R&D Program of China (2022YFC2303200, 2022YFC2303502), Strategic Priority Research Program of the Chinese Academy of Sciences (XDB0940102),Three-Year Initiative Plan for Strengthening Public Health System Construction in Shanghai (2023–2025) Key Discipline Project (GWVI-11.1-09), Shanghai Municipal Science and Technology Major Project (2019SHZDZX02), State Key Laboratory of Respiratory Disease Project (J24411029), Guangdong Basic and Applied Basic Research Foundation (2023A1515010152), Young

Scientists Fund of the Guangzhou National Laboratory (QNPG23-03), and China Postdoctoral Science Foundation (2024M763413).

## Additional information

### Funding

| Funder | Grant reference number | Author |
|---|---|---|
| National Natural Science Foundation of China | 92269202 | Guangxun Meng |
| National Natural Science Foundation of China | 82825001 | Jincun Zhao |
| National Natural Science Foundation of China | 92054104 | Yaming Jiu |
| National Natural Science Foundation of China | 82402593 | Shi Yu |
| National Key Research and Development Program of China | 2022YFC2303200 | Guangxun Meng |
| National Key Research and Development Program of China | 2022YFC2303502 | Yaming Jiu |
| Chinese Academy of Sciences | Strategic Priority Research Program XDB0940102 | Guangxun Meng |
| Three-Year Initiative Plan for Strengthening Public Health System Construction in Shanghai (2023-2025) Key Discipline Project | GWVI-11.1-09 | Min Chen |
| Shanghai Municipal Science and Technology Major Project | 2019SHZDZX02 | Guangxun Meng |
| State Key Laboratory of Respiratory Disease | J24411029 | Shi Yu |
| Guangdong Basic and Applied Basic Research Foundation | 2023A1515010152 | Shi Yu |
| Young Scientist Fund of the Guangzhou National Laboratory | QNPG23-03 | Shi Yu |
| China Postdoctoral Science Foundation | 2024M763413 | Xu Zheng |

The funders had no role in study design, data collection and interpretation, or the decision to submit the work for publication.

### Author contributions

Xu Zheng, Conceptualization, Data curation, Formal analysis, Funding acquisition, Investigation, Methodology, Writing – original draft, Writing – review and editing; Shi Yu, Conceptualization, Investigation, Methodology, Writing – original draft, Writing – review and editing; Yanqiu Zhou, Funding acquisition, Investigation; Kuai Yu, Yuhui Gao, Mengdan Chen, Dong Duan, Yunyi Li, Xiaoxian Cui, Jiabin Mou, Yuying Yang, Xun Wang, Investigation; Min Chen, Resources, Funding acquisition; Yaming Jiu, Resources, Funding acquisition, Investigation, Methodology, Writing – review and editing; Jincun Zhao, Resources, Funding acquisition, Investigation; Guangxun Meng, Conceptualization, Resources, Supervision, Funding acquisition, Methodology, Writing – review and editing

## Author ORCIDs

Xu Zheng ⓘ https://orcid.org/0000-0002-1446-1882
Shi Yu ⓘ https://orcid.org/0000-0003-0651-4769
Kuai Yu ⓘ https://orcid.org/0000-0001-9699-4780
Guangxun Meng ⓘ https://orcid.org/0000-0003-4634-4661

## Ethics

Animal experiments related to authentic SARS-CoV-2 were conducted in Guangzhou Customs District Technology Center BSL-3 Laboratory (Permit Number: IQTC202302).

Reviewer #1 (Public review): https://doi.org/10.7554/eLife.98593.3.sa1
Reviewer #2 (Public review): https://doi.org/10.7554/eLife.98593.3.sa2
Author response https://doi.org/10.7554/eLife.98593.3.sa3

---

## Additional files

### Supplementary files

Supplementary file 1. Primer sequences for RT-qPCR.

Supplementary file 2. Primer sequences for sgRNA.

MDAR checklist

### Data availability

All data generated or analysed during this study are included in the manuscript and supporting files; source data files have been provided for figures and figure supplements.

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
