## [Editor Report · eLife Assessment]

This study provides **important** insights into how IL-1 cytokines protect cells against SARS-CoV-2 infection. By inducing a non-canonical RhoA/ROCK signaling pathway, IL-1beta inhibits the ability of SARS-CoV-2 infected cells to fuse with uninfected cells and produce syncytia. **Convincing** evidence underlies the identification of the key signaling components required for this inhibitory phenotype, and suggests that this process may also function to inhibit SARS-CoV-2 infection in vivo.

---

## [Referee Report · Reviewer #1 (Public review)]

Summary:

SARS-CoV-2 infection induces syncytia formation, which promotes viral transmission. In this paper, the authors aimed to understand how host-derived inflammatory cytokines IL-1α/β combat SARS-CoV-2 infection.

Strengths:

First, they used a cell-cell fusion assay developed previously to identify IL-1α/β as the cytokines that inhibit syncytia formation. They co-cultured cells expressing the spike protein and cells expressing ACE2 and found that IL-1β treatment decreased syncytia formation and S2 cleavage.

Second, they investigated the IL-1 signaling pathway in detail, using knockouts or pharmacological perturbation to understand the signaling proteins responsible for blocking cell fusion. They found that IL-1 prevents cell-cell fusion through MyD88/IRAK/TRAF6 but not TAK1/IKK/NF-κB, as only knocking out MyD88/IRAK/TRAF6 eliminates the inhibitory effect on cell-cell fusion in response to IL-1β. This revealed that the inhibition of cell fusion did not require a transcriptional response and was mediated by IL-1R proximal signaling effectors.

Third, the authors identified RhoA/ROCK activation by IL-1 as the basis for this inhibition of cell fusion. By visualizing a RhoA biosensor and actin, they found a redistribution of RhoA to the cell periphery and cell-cell junctions after IL-1 stimulation. This triggered the formation of actin bundles at cell-cell junctions, preventing fusion and syncytia formation. The authors confirmed this molecular mechanism by using constitutively active RhoA and an inhibitor of ROCK.

Diverse Cell types and in vivo models were used, and consistent results were shown across diverse models. These results were convincing and well-presented.

In summary, the authors have provided compelling evidence regarding how IL-1 signaling induces a prophylactic response to viral infection. While the mechanistic details of how IL-1R and MyD88 induce RhoA/Rock pathway to mediate actin remodeling remain unclear, this manuscript serves as the basis for future studies.

---

## [Referee Report · Reviewer #2 (Public review)]

Summary:

In this study, Zheng et al investigated the role of inflammatory cytokines in protecting cells against SARS-CoV-2 infection. They demonstrate that soluble factors in the supernatants of TLR stimulated THP1 cells reduce fusion events between HEK293 cells expressing SARS-CoV-2 S protein and the ACE2 receptor. Using qRT-PCR and ELISA, they demonstrate that IL-1 cytokines are (not surprisingly) upregulated by TLR treatment in THP1 cells. Further, they convincingly demonstrate that recombinant IL-1 cytokines are sufficient to reduce cell-to-cell fusion mediated by the S protein. Using chemical inhibitors and CRISPR knock-out of key IL-1 receptor signaling components in HEK293 cells, they demonstrate that components of the myddosome (MYD88, IRAK1/4, and TRAF6) are required for fusion inhibition, but that downstream canonical signaling (i.e., TAK1 and NFKB activation) is not required. Instead, they provide evidence that IL-1-dependent non-canonical activation of RhoA/Rock is important for this phenotype. Importantly, the authors demonstrate that expression of a constitutively active RhoA alone is sufficient to inhibit fusion and that chemical inhibition of Rock could reverse this inhibition. The authors followed up these in vitro experiments by examining the effects of IL-1 on SARS-COV-2 infection in vivo and they demonstrate that recombinant IL-1 can reduce viral burden and lung pathogenesis in a mouse model of infection. Use of a ROCK inhibitor in IL-1 treated mice restored the ability of SARS-CoV-2 to spread in the lung, suggesting that this inhibitory process functions in vivo.

Strengths:

(1) The bioluminescence cell-cell fusion assay provides a robust quantitative method to examine cytokine effects on viral glycoprotein-mediated fusion.

(2) The study identifies a new mechanism by which IL-1 cytokines can limit virus infection.

(3) The authors tested IL-1 mediated inhibition of fusion induced by many different coronavirus S proteins and several SARS-CoV-2 strains.

(4) The authors demonstrate that recombinant IL-1 mediated inhibition of SARS-CoV-2 infection in mice is dependent on the RhoA/Rock pathway.

---

## [Author Response]

The following is the authors’ response to the original reviews.

**Public Reviews:**

**Reviewer #1 (Public Review):**
Summary:SARS-CoV-2 infection induces syncytia formation, which promotes viral transmission. In this paper, the authors aimed to understand how host-derived inflammatory cytokines IL-1α/β combat SARS-CoV-2 infection.Strengths:First, they used a cell-cell fusion assay developed previously to identify IL-1α/β as the cytokines that inhibit syncytia formation. They co-cultured cells expressing the spike protein and cells expressing ACE2 and found that IL-1β treatment decreased syncytia formation and S2' cleavage.Second, they investigated the IL-1 signaling pathway in detail, using knockouts or pharmacological perturbation to understand the signaling proteins responsible for blocking cell fusion. They found that IL-1 prevents cell-cell fusion through MyD88/IRAK/TRAF6 but not TAK1/IKK/NF-κB, as only knocking out MyD88/IRAK/TRAF6 eliminates the inhibitory effect on cell-cell fusion in response to IL-1β. This revealed that the inhibition of cell fusion did not require a transcriptional response and was mediated by IL-1R proximal signaling effectors.Third, the authors identified RhoA/ROCK activation by IL-1 as the basis for this inhibition of cell fusion. By visualizing a RhoA biosensor and actin, they found a redistribution of RhoA to the cell periphery and cell-cell junctions after IL-1 stimulation. This triggered the formation of actin bundles at cell-cell junctions, preventing fusion and syncytia formation. The authors confirmed this molecular mechanism by using constitutively active RhoA and an inhibitor of ROCK.Diverse Cell types and in vivo models were used, and consistent results were shown across diverse models. These results were convincing and well-presented.Weaknesses:As the authors point out in the discussion, whether IL-1-mediated RhoA activation is specific to viral infection or regulates other RhoA-regulated processes is unclear. We would also require high-magnification images of the subcellular organization of the cytoskeleton to appreciate the effect of IL-1 stimulation.

Thanks for the suggestions. We tested the role of IL-1β in other RhoA-regulated processes, and found that IL-1β-mediated RhoA activation also reduced cell migration in a cell scratch assay (see Author response image 1). We also provided high-magnification images in the revised Figures 4 and 5, as well as their respective figure supplements.

(A) Cell scratch assay images of HEK293T cells treated with PBS or IL-1β. (B) Quantification of cell migration in (A).

**Reviewer #2 (Public Review):**
Summary:In this study, Zheng et al investigated the role of inflammatory cytokines in protecting cells against SARS-CoV-2 infection. They demonstrate that soluble factors in the supernatants of TLR-stimulated THP1 cells reduce fusion events between HEK293 cells expressing SARS-CoV-2 S protein and the ACE2 receptor. Using qRT-PCR and ELISA, they demonstrate that IL-1 cytokines are (not surprisingly) upregulated by TLR treatment in THP1 cells. Further, they convincingly demonstrate that recombinant IL-1 cytokines are sufficient to reduce cell-to-cell fusion mediated by the S protein. Using chemical inhibitors and CRISPR knock-out of key IL-1 receptor signaling components in HEK293 cells, they demonstrate that components of the myddosome (MYD88, IRAK1/4, and TRAF6) are required for fusion inhibition, but that downstream canonical signaling (i.e., TAK1 and NFKB activation) is not required. Instead, they provide evidence that IL-1-dependent non-canonical activation of RhoA/Rock is important for this phenotype. Importantly, the authors demonstrate that expression of a constitutively active RhoA alone is sufficient to inhibit fusion and that chemical inhibition of Rock could reverse this inhibition. The authors followed up these in vitro experiments by examining the effects of IL-1 on SARS-COV-2 infection in vivo and they demonstrate that recombinant IL-1 can reduce viral burden and lung pathogenesis in a mouse model of infection. However, the contribution of the RhoA/Rock pathway and inhibition of fusion to IL-1-mediated control of SARS-CoV-2 infection in vivo remains unclear.Strengths:(1) The bioluminescence cell-cell fusion assay provides a robust quantitative method to examine cytokine effects on viral glycoprotein-mediated fusion.(2) The study identifies a new mechanism by which IL-1 cytokines can limit virus infection.(3) The authors tested IL-1 mediated inhibition of fusion induced by many different coronavirus S proteins and several SARS-CoV-2 strains.Weaknesses:(1) The qualitative assay demonstrating S2 cleavage and IL-1 mediated inhibition of this phenotype is extremely variable across the data figures. Sometimes it appears like S2 cleavage (S2') is reduced, while in other figures immunoblots show that total S2 protein is decreased. Based on the proposed model the expectation would be that S2 abundance would be rescued when cleavage is inhibited.

In our present manuscript, IL-1-mediated changes of the full-length spike showed some variation between authentic SARS-CoV-2 infection model and HEK293T-S + HEK293T-ACE2 coculture model, while IL-1 inhibited S2’ cleavage accompanied by a reduction of S2 subunit in both models.

In the authentic SARS-CoV-2 infection model, we observed that IL-1 inhibited S2' cleavage accompanied with a reduction in both S2 subunit and full-length spike protein. This is likely because the S2 subunit and full-length spike protein in this model are not only from infected cells, but also from intracellular viral particles. IL-1 inhibited SARS-CoV-2 induced cell-cell fusion and reduced the viral load in host cells, therefore the abundance of S2 subunit and full-length spike proteins were both reduced.

In the HEK293T-based co-culture model, IL-1 inhibited S2' cleavage accompanied with a reduction in S2 subunit, while the full-length spike protein was more or less rescued. Based on our previous study, R685A and ΔRRAR spike mutants cannot generate the S2 subunit, but still generated S2′ fragment to induce cell-cell fusion, and the S2' fragment produced from R685A and ΔRRAR spike mutants were only slightly reduced compared to wild-type spike protein, suggesting that the S2' fragment is mainly derived from the full-length spike directly, and to a minimal extent from the S2 subunit (Fig. 4B and 4G, PMID: 34930824). Thus, inhibition of S2’ cleavage by IL-1 mainly rescued the full-length spike protein.

(2) The text referencing Figure 1H suggests that TLR-stimulated THP-1 cell supernatants "significantly" reduce syncytia, but image quantification and statistics are not provided to support this statement.

Thanks for pointing out this issue. We have provided fluorescence image quantification and statistics in the revised version of our manuscript (Figure 1D, Figure 1-figure supplement 1A, Figure 1H-1I, Figure 2H-2I, Figure 1-figure supplement 1D-1E, Figure 1-figure supplement 1H-1I, Figure 2-figure supplement 1C-1D, Figure 2-figure supplement 2B-2E, Figure 2-figure supplement 2G-2H, Figure 2-figure supplement 6A-6B, Figure 2-figure supplement 7F-7G).

(3) The authors conclude that because IL-1 accumulates in TLR2-stimulated THP1 monocyte supernatants, this cytokine accounts for the ability of these supernatants to inhibit cell-cell fusion. However, they do not directly test whether IL-1 is required for the phenotype. Inhibition of the IL-1 receptor in supernatant-treated cells would help support their conclusion.

Thanks for the suggestion. Accordingly, we performed experiment and found that IL-1RA treatment reduced the inhibitory effect of PGN-stimulated THP-1 cell culture supernatant on cell-cell fusion, suggesting that IL-1 is required for the inhibition. This result has been added in our revised manuscript (Figure 2J and Figure2-figure supplement 4C).

(4) Immunoblot analysis of IL-1 treated HEK293 cells suggests that this cytokine does not reduce the abundance of ACE2 or total S protein in cells. However, it is possible that IL-1 signaling reduces the abundance of these proteins on the cell surface, which would result in a similar inhibition of cell-cell fusion. The authors should confirm that IL-1 treatment of their cells does not change Ace2 or S protein on the cell surface.

Thanks for the suggestion. Accordingly, we applied Wheat Germ Agglutinin (WGA) to stain cell surface in HEK293T cells and observed that IL-1β treatment did not change ACE2 or Spike protein on the cell surface. This result has been added in our revised manuscript (Figure 5-figure supplement 3A-D).

(5) In Figure 5A, expression of constitutively active RhoA appears to have profound effects on how ACE2 runs by SDS-PAGE, suggesting that RhoA may have additional effects on ACE2 biology that might account for the decreased cell-cell fusion. This phenotype should be addressed in the text and explored in more detail.

Thanks for pointing out this. We also noticed that the occurrence of cell-cell fusion reduced the amount of ACE2, whereas inhibition of cell-cell fusion restored the ACE2 abundance. Take the original Figure 5A (revised Figure 4-figure supplement 2B) as example, the increased ACE2 protein should be attributed to the decreased cell-cell fusion upon RhoA-CA transfection, as Spike binding with ACE2 leads to clathrin- and AP2-dependent endocytosis, resulting in ACE2 degradation in the lysosome (PMID: 36287912).

In addition, we have examined the potential effect of RhoA-CA on ACE2, and found that RhoA-CA did not affect ACE2 expression, nor Spike binding to ACE2 (revised Figure 5-figure supplement 2E); it did not affect ACE2 distribution on cell surface either (revised Figure 5-figure supplement 2F and G).

(6) The experiments linking IL-1 mediated restriction of SARS-COV-2 fusion to the control of virus infection in vivo are incomplete. The reported data demonstrate that recombinant IL-1 can restrict virus replication in vivo, but they fall short of confirming that the in vitro mechanism described (reduced fusion) contributes to the control of SARS-CoV2 replication in vivo. A critical piece of data that is missing is the demonstration that the ROCK inhibitor phenocopies IL-1RA treatment of SARS-COV-2 infected mice (viral infection and pathology).

Thanks for this suggestion. Accordingly, we applied the ROCK inhibitor in vivo to confirm its role in SARS-CoV-2-infected mice, and found similar phenotype as the IL-1RA treatment experiment. That is to say, Y-26732 treatment prevented the formation of IL-1β-induced actin bundles at cell-cell junctions, thus promoted syncytia formation and further viral transmission in vivo (revised Figure 7).

**Recommendations for the authors:**

**Reviewer #1 (Recommendations For The Authors):**
I suggest providing single-channel images in a supplementary figure for the live-cell images in Figures 4 and 5. Higher magnification images would also help distinguish the subcellular details of the cytoskeleton organization.

Thanks for the suggestion. We have provided the single channel images and higher magnification images in the revised Figures 4 and 5, as well as their respective figure supplements.

In Figure 4, the authors showed that IL-1 activates RhoA and induces the accumulation of activated RhoA at the cell-cell junctions. They also showed that IL-1 promotes the formation of actin bundles at cell-cell junctions. However, the authors have not shown any connection between RhoA and actin yet, but in lines 263-264, they claim that actin bundle formation is induced by RhoA. Evidence for this part was shown in later results, but at this moment, it is lacking. The same applies to lines 282-284; I think this conclusion that IL-1-induced actin bundle formation is through the RhoA-ROCK pathway should come after showing how RhoA affects actin bundle formation at cell-cell junctions. To this end, I suggest moving Supplementary Figures S12B and S12D to the main figure, as they provide strong evidence of the IL-1-RhoA-ROCK-actin pathway.

We appreciate these valuable comments. As suggested, we have moved the respective supplementary figures to the main figures to support our findings in the revised manuscript (Figure 4E and Figure 4-figure supplement 2B; Figure 5C and Figure 5-figure supplement 2A), the text has also been adjusted accordingly.